# Towards robust vision by multi-task learning on monkey visual cortex

**Shahd Safarani,[1,*] Arne Nix,[1-2] Konstantin Willeke,[1-2] Santiago A. Cadena,[2-3]**
**Kelli Restivo,[4-5] George Denfield,[6] Andreas S. Tolias,[4-5] Fabian H. Sinz[1-5,**]**

[1] Institute for Bioinformatics and Medical Informatics, University Tübingen, Germany
[2] International Max Planck Research School for Intelligent Systems, Tübingen, Germany
[3] Institute for Computer Science, University of Göttingen, Germany
[4] Department for Neuroscience, Baylor College of Medicine, Houston, TX, USA
[5] Center for Neuroscience and Artificial Intelligence, Baylor College of Medicine, Houston, TX, USA
[6] Columbia University, Department of Psychiatry, New York, USA

[*]shahdsaf@hotmail.com, [**]sinz@cs.uni-goettingen.de

## Abstract

Deep neural networks set the state-of-the-art across many tasks in computer vision, but their generalization ability to simple image distortions is surprisingly fragile. In contrast, the mammalian visual system is robust to a wide range of perturbations. Recent work suggests that this generalization ability can be explained by useful inductive biases encoded in the representations of visual stimuli throughout the visual cortex. Here, we successfully leveraged these inductive biases with a multi-task learning approach: we jointly trained a deep network to perform image classification and to predict neural activity in macaque primary visual cortex (V1) in response to the same natural stimuli. We measured the out-of-distribution generalization abilities of our resulting network by testing its robustness to common image distortions. We found that co-training on monkey V1 data indeed leads to increased robustness despite the absence of those distortions during training. Additionally, we showed that our network's robustness is often very close to that of an Oracle network where parts of the architecture are directly trained on noisy images. Our results also demonstrated that the network's representations become more brain-like as their robustness improves. Using a novel constrained reconstruction analysis, we investigated what makes our brain-regularized network more robust. We found that our monkey co-trained network is more sensitive to content than noise when compared to a Baseline network that we trained for image classification alone. Using DeepGaze-predicted saliency maps for ImageNet images, we found that the monkey co-trained network tends to be more sensitive to salient regions in a scene, reminiscent of existing theories on the role of V1 in the detection of object borders and bottom-up saliency. Overall, our work expands the promising research avenue of transferring inductive biases from biological to artificial neural networks on the representational level, and provides a novel analysis of the effects of our transfer.

## 1   Introduction

Although machine learning algorithms have witnessed enormous progress thanks to the recent success of deep learning methods [1], current state-of-the-art deep models [2–4] still fall behind the generalization abilities of biological brains [5, 6]. This includes a lack of robustness to image

35th Conference on Neural Information Processing Systems (NeurIPS 2021).

corruptions as pointed out by Hendrycks and Dietterich [7], who measured a network's performance on 15 different image corruptions applied to the ImageNet [8] test-set. Studies with similar image corruptions have proven to severely decrease performance in classification networks while having a smaller impact on human perception [9], suggesting that the ability to *extrapolate* is weak in these networks compared to the mammalian visual system. This gap in extrapolation has previously been attributed to differences in feature representations [10, 11] and internal strategies for decision making [12] between humans and CNNs.

Historically, neuroscience has inspired many innovations in artificial intelligence [13, 14], and most of this transfer between neuroscience and machine learning happens on the implementational level [15, 13]. However, we currently know too little about the structure of the brain at the level of detail needed to transfer functional generalization properties from biological to artificial systems [5]. To transfer functional inductive biases from the brain to deep neural networks (DNNs), it may thus be better to consider the representational level by capturing biological feature representations in the responses of biological neurons to visual input – abstracting away from the implementational level. In fact, deep neural networks have set new standards in capturing brain activity across multiple areas in the visual cortex [16], becoming the state-of-the-art for neural response prediction of the primary visual cortex (area V1, [17], but also see Marques et al. [18] for biologically inspired models that perform similarly well). Recent work has shown that CNNs, which were fitted to V1 neural data, can generalize well to other neurons, animals, and stimuli [19, 20]. Vice versa, prior work suggests that enforcing brain-like representations in CNNs via neural data from humans [21], mice [22], or monkeys [23] can have beneficial effects on the generalization abilities of these networks to stimuli outside their training distribution for object recognition.

Our work expands on this line of research by exploring the extrapolation capabilities of multi-task learning models (MTL; [24]) trained on image classification and prediction of neural responses from monkey V1. This approach was proposed as the *neural co-training hypothesis* by Sinz et al. [5], but remained untested to the best of our knowledge. We implement MTL via a shared representation between image classification and neural response prediction (Fig. 1). Our hypothesis is that MTL with neural data regularizes the shared representation to inherit good functional inductive biases from neural data, and improves the network's generalization abilities to out-of-distribution images, thus rendering it more robust. We empirically test this idea using common corruptions on tiny ImageNet (TIN)[1]. We show that MTL on monkey V1 has a positive effect on generalization as it increases the model's robustness to image distortions, even though it is trained on undistorted images only. We compare our model with a robust Oracle model to quantify what performance improvement can be expected given that only parts of the network are shared during MTL. Subsequently, we develop a constrained reconstruction based method to analyze learned sensitivities and invariances in the feature representation of the different models. We find that the robust models qualitatively exhibit different feature sensitivities than a standard classification model. Finally, we show that the feature sensitivity of the monkey V1 co-trained model is related to salient image features, consistent with existing theories about the role of V1 [25]. Overall, our results support the neural co-training hypothesis and further expand the scope of prior results, by exploring the relationship between brain-like representations and robustness. To the best of our knowledge, we are the first to analyze learned feature representations of neurally co-trained models and thus take a step towards a semantic understanding of what makes biological vision robust.

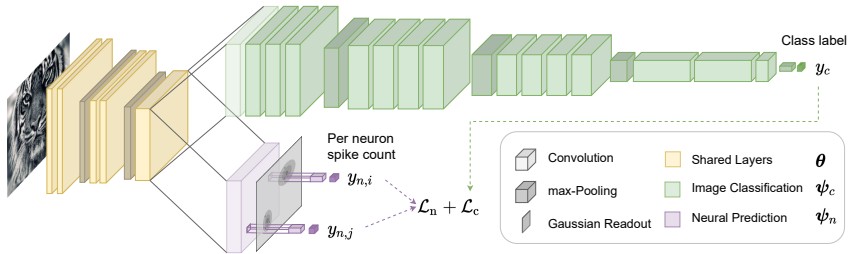

Figure 1: VGG-19 architecture for MTL on image classification and neural prediction.

## 2   Neural multi-task learning

**Data**   The images for the classification task and the neurophysiological experiments were selected from the ImageNet dataset [8]. For the classification task, we used a grayscale version of TIN[1]. The tiny ImageNet dataset (TIN) is a subset of ImageNet containing 100000 training images of 200 classes (500 for each class) downsized to images of size 64x64. In addition, each class has 50 validation images and 50 test images. For neural prediction, we used neurophysiological recordings of 458 neurons from the primary visual cortices (area V1) of two fixating awake macaque monkeys, recorded with a 32-channel depth electrode during 15 (monkey 1) and 17 (monkey 2) sessions. In each session, approximately 1000 trials of 15 images were presented – each image for 120ms. We extracted the spike count from 40ms to 160ms after image onset. The image set presented to the monkey consists of 24075 images from 964 categories – 25 images per category. Of those, 24000 were designated to model training and 75 to testing. For each training trial, a new subset of 15 images was randomly sampled from the training set. Test images were displayed in fixed order in 5 test trials consisting of 15 images each. Each of the test trials were randomly interleaved among training trials and repeated 40-50 times per session. All images were converted to gray-scale and presented at $420{\times}420$ px, covering 6.7° visual angle for the monkey, resulting in 63 pixels per degree (ppd). For model training, images were downscaled and cropped to $64{\times}64$ pixels, corresponding to 14.0 ppd.

**Architecture**   All our experiments were based on a variant of the VGG-19 architecture [26] with batch normalization layers [27] after every convolutional layer (Figure 1). To allow for arbitrary image sizes, we made the network fully convolutional by replacing the fully connected readout by three convolutional layers with dropout of 0.5 after the first two, and a final max-pooling operation and softmax [28]. We predicted neural responses by feeding the output of the convolutional layer `conv-3-1`, shown by Cadena et al. [17] to be optimal for predicting V1 responses, into a *Gaussian readout* [19] yielding a spike count prediction per neuron and image.

**Models**   We use a VGG-19, like we described it above, trained on grayscale TIN to serve as the *Baseline* for image classification in our experiments. To prepare our neural co-training, similar to Li et al. [22], we first trained a *Monkey Predictor* model on the image-response pairs of our recorded neural data. We then used that model to predict neural responses for all input images of the TIN classification dataset. These predicted responses served as the basis neural dataset we used in our MTL approach. This allowed us to balance the amount of data we have for each task and it removed trial-to-trial variability in the neural data.

Since co-training only affects the shared representation up to layer `conv-3-1`, we cannot expect the network to be as robust as a network where all layers are trained on data augmented with the image distortions. To explore the limits on robustness resulting from sharing lower layers only, we trained a classification model with a 1:1 mixture of clean and distorted images drawn from the pool of 14 ImageNet-C [7] corruptions (cf. Figure 9 for examples). To push the robustness to the frozen part, we added a second loss that penalizes the Euclidean distance between the outputs of layer `conv-3-1` for the same image augmented with different corruptions – similar to Chen et al. [29]. We then froze all layers up to `conv-3-1`, re-initialized the rest, and re-trained the remaining network on clean data only. We refer to this model as the *Oracle* since it has access to the image distortions during training – unlike our MTL models.

To demonstrate that MTL can in principle transfer robustness properties without showing distorted images in training, we generated neural responses from our Oracle model for all images of the *clean* TIN dataset by freezing the Oracle model and training a Gaussian readout on top of layer `conv-3-1` for 10 epochs to predict V1 data. Then, we trained a model on the resulting neural responses alongside clean image classification using MTL. We call this model *MTL-Oracle*. This model also gives us a realistic "upper bound" for our MTL experiments.

For our main experiment, we trained MTL with the neural responses generated from the Monkey Predictor model, and refer to it as *MTL-Monkey*. This model has never seen distorted images at any point during training. To demonstrate that MTL-Monkey has an effect beyond introducing noise into the training, we perform a control experiment which we refer to as *MTL-Shuffled*. For this, we train a model on the same neural data but with shuffled responses across images for all neurons. An overview of all models used in this study can be found in Table 1.

---

[1]https://www.kaggle.com/c/tiny-imagenet/overview

Table 1: Overview of the different models that we use in this study.

| Model | Classification | Neural Prediction |
|---|---|---|
| ■ Baseline | Clean TIN | – |
| Monkey Predictor | – | Monkey responses |
| ■ Oracle | Noise augmented TIN | – |
| ■ MTL-Oracle | Clean TIN | Oracle model responses |
| ■ MTL-Monkey | Clean TIN | Monkey predictor responses |
| ■ MTL-Shuffled | Clean TIN | Monkey predictor responses (shuffled) |

**Training** We used cross-entropy loss for single task *image classification* and Poisson loss for single task *neural prediction*. For *multi-task training*, the challenge was to find the optimal balance between the two objectives that achieves a reasonable performance on each task individually, and allows both tasks to benefit from each other by learning common representations. To put both objectives on the same scale, we used their corresponding negative log-likelihood and learned their balance through trainable observation noise parameters $\sigma$ [30]. This yields a combined loss of $\frac{1}{\sigma_c^2}\mathcal{L}_{\text{CE}}(\boldsymbol{\theta}, \boldsymbol{\psi}_c) + \frac{1}{2\sigma_n^2}\mathcal{L}_{\text{MSE}}(\boldsymbol{\theta}, \boldsymbol{\psi}_n) + \log\sigma_c + \log\sigma_n$ where $\boldsymbol{\theta}$ are the shared parameters and $\boldsymbol{\psi}_c, \sigma_c$ and $\boldsymbol{\psi}_n, \sigma_n$ are the task-specific parameters for $c$lassification and $n$eural prediction, respectively. The classification objective $\mathcal{L}_{\text{CE}}$ was the standard cross-entropy, analogous to the single-task case. For MTL on neural data, we used mean-squared error $\mathcal{L}_{\text{MSE}}$ because the targets are predictions from the network trained on neural data and not the original noisy neural responses. For optimization, we accumulated the gradients over the different losses to optimize the shared parameters $\boldsymbol{\theta}$ in a single combined gradient step. By definition, the two loss components would contribute equally to the learning process.

We standardized all pixel values with the mean and standard deviation of the training set, and augmented the images by random cropping, horizontal flipping, and rotations in a range of $15°$ for classification. Optimization was performed using stochastic gradient descent with momentum in all classification-related cases, and Adam for single task neural prediction [31]. We used a batch-size of 128 and weight decay with a factor of $5 \cdot 10^{-4}$ throughout all our experiments. The initial learning rate was determined for each task individually and reduced by a (task-specific) factor via an adaptive learning rate schedule. The schedule reduces the learning rate depending on the validation performance – classification performance in the case of MTL – when the rate of improvement is not above $10^{-4}$ for 5 consecutive epochs. The training was stopped when we reach either five learning rate reduction steps or a maximum number of epochs, that we defined for each task. This training setup was determined via prior hyper-parameter searches on the validation-set. We repeated every experiment with five different random initializations. Error bars were obtained by bootstrapping (250 repetitions).

## 3 Results

Our goal is to test whether co-training on neural data can lead to improved extrapolation abilities. To this end, we evaluated our model's robustness on distorted copies of the TIN validation-set – used as a test-set in our experiments – following the corruption paradigm of Hendrycks and Dietterich [7]. We reproduced the distortions with an on-the-fly implementation [32], dropped *glass blur* because it is computationally expensive, and refer to our resulting test-set as *TIN-TC*. We quantified the robustness for each of the remaining 14 noise types and five levels of corruption severity separately, and computed a summary robustness score adopted from Hendrycks and Dietterich [7]: $\frac{1}{14}\sum_{c=1}^{14}\overline{A}_c^{\text{robust}}/\overline{A}_c^{\text{Baseline}}$, where $\overline{A}_c = \frac{1}{5}\sum_{l,s=1}^{5}A_{l,c,s}$ denotes the mean accuracy on corruption $c$ across levels $l$ and seeds $s$.

**MTL can successfully transfer robustness** Comparing the robustness of the MTL-Oracle model on TIN-TC to the robustness of the single-task *Baseline* model trained on clean TIN only, we saw clear signs of successful transfer (Fig. 2 and Fig. 3A,B) although the MTL network has never seen the image distortions of TIN-TC. In fact, the MTL-Oracle performed close to the Oracle in most cases.

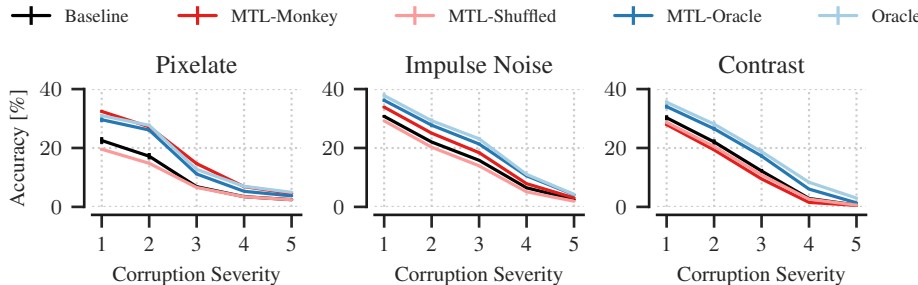

Figure 2: Exemplary classification results on TIN-TC, showing 3 corruption types with the best (left), median (center) and worst (right) robustness scores for MTL-Monkey across 5 increasing levels of severity each.

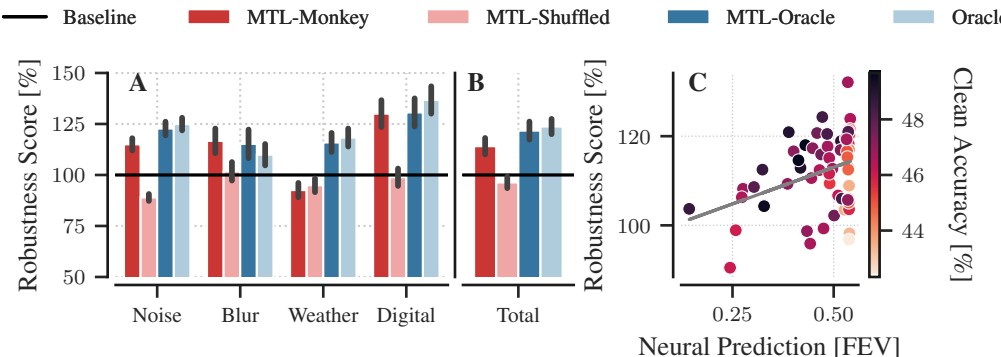

Figure 3: **A** Robustness scores for each model grouped by corruption category, as defined in Hendrycks and Dietterich [7]. **B** Overall robustness scores for our 5 different models. **C** Robustness and neural prediction correlate positively for MTL-Monkey models across 12 different batch-ratios and 5 random seeds per model (grey line: linear regression from neural performance to robustness). Neural prediction performance is measured as the fraction of explained variance (FEV), as described in Cadena et al. [17]. A darker color indicates higher accuracy on the clean TIN test-set.

**Co-training with monkey V1 increases robustness.** The results of MTL-Oracle show that MTL with neural responses from a robust network in response to undistorted images successfully transfers robustness properties. Furthermore, MTL-Monkey generalized better to the TIN-TC image distortions than the Baseline model, similar to MTL-Oracle, despite the fact that MTL-Monkey has not seen distorted images at any stage during the training process. We found increased robustness for 9/14 image corruptions. This improvement is mainly observed across 3 groups of distortions: *Noise*, *Blur* and *Digital* (Fig. 3A), whereas MTL-Monkey did not exceed the Baseline performance for the *Weather* group. The shuffled control did not provide any benefits (Fig. 2 and Fig. 3A,B), suggesting that the improved robustness of our monkey network is, in fact, due to the original neural data.

**The more "brain-like" the neural network, the better it generalizes to image distortions.** If features in the neural data affect the robustness, we would expect that the robustness of MTL-Monkey correlates positively with its neural prediction performance on real monkey V1 data. To test this hypothesis, we created a pool of MTL-Monkey models with varying neural performance by altering the amount of neural data introduced during co-training. We ran experiments with different ratios of neural data to image classification that was presented to the network before each backward pass. We ran experiments for ratios $1:15, 1:10, 1:7, 1:5, 1:4, 1:3, 1:2, 1:1, 2:1, 3:1, 4:1$, and $5:1$ with five seeds each, giving us 60 models to plot in Figure 3-C. We found that both the model's test accuracy on clean images and its neural performance on real monkey V1 data improved the network's robustness (Figure 3C; $p < 10^{-4}$ for both neural prediction and clean accuracy[2]).

---

[2]t-test for both factors in a 2-factor linear regression, in which robustness (dependent variable) is predicted from clean test accuracy for image classification and performance on V1 prediction (independent variables).

Analysis for MTL-Shuffled showed a slight connection between robustness and neural performance ($p = 0.034$ for neural prediction and $p < 10^{-13}$ for clean accuracy). When comparing the regression coefficient of neural prediction in the case of MTL-Monkey and MTL-Shuffled, we found that the influence of neural performance on robustness is two orders of magnitude larger for real neural data $b_{monkey} = 54.72$ than for the shuffled version $b_{shuffled} = 0.26$. Overall, our results are consistent with previous work finding a positive correlation between model robustness and "brain-likeness" [33].

## 4   Analysis

In the previous section, we showed that our MTL approach can transfer robustness properties and that improved robustness correlates with more brain-like representations learned from monkey V1 data. The aim of this section is to understand the representational differences compared to other models that could be responsible for the increased robustness of our MTL-Monkey model. To this end, we visualize which image features the networks are sensitive to, using a novel resource constrained image reconstruction from a given layer across all of the models used in this study. The rationale behind a constrained reconstruction is to put a resource limitation on the total power of an image, thereby force the reconstruction to put contrast in the image wherever it is necessary to recreate the activity of a given layer, and thus visualize the sensitivities and invariances of this layer (see Figure 4). Specifically, given a noisy target image $\mathbf{x}_0$, we computed the corresponding activations $f(\mathbf{x}_0)$ of a particular layer and reconstructed the original image by minimizing the squared loss between the target activations and the activations from the reconstructed image $\ell(\mathbf{x}_0, \mathbf{x}) = \|f(\mathbf{x}) - f(\mathbf{x}_0)\|^2$ subject to a norm constraint

$$\mathbf{x}^* = \text{argmin}_{\mathbf{x}} \|f(\mathbf{x}) - f(\mathbf{x}_0)\|^2 \text{ s.t.} \|\mathbf{x}\|^2 \leq r^2.$$

Note that, if $\|\mathbf{x}_0\| \leq r$, one trivial optimal solution is $\mathbf{x} = \mathbf{x}_0$. However, if $\|\mathbf{x}_0\| > r$, the constraint becomes active and the reconstruction has to choose where to put power in the image (see Figure 4). This can be seen more formally if we approximate the loss function with a second order Taylor approximation $\ell(\mathbf{x}_0, \mathbf{x}) \approx \frac{1}{2}(\mathbf{x} - \mathbf{x}_0)^\top H(\mathbf{x} - \mathbf{x}_0)$ around the optimal solution $\mathbf{x}_0$. Using the local approximation in the optimization problem and solving for $\mathbf{x}$ using Lagrange multipliers

$$\text{max}_\gamma \text{ min}_{\mathbf{x}} \frac{1}{2}(\mathbf{x} - \mathbf{x}_0)^\top H(\mathbf{x} - \mathbf{x}_0) + \gamma \cdot \frac{1}{2}\left(\|\mathbf{x}\|_2^2 - r^2\right) \text{ s.t. } \gamma \geq 0$$

yields $\mathbf{x} = (H + \gamma I)^{-1} H \mathbf{x}_0$ where $\gamma$ denotes the Lagrange multiplier chosen such that $\|\mathbf{x}^*\| = r$ if the constraint is active. To see that this preferentially reconstructs images along directions where the loss in activation space is more sensitive, consider $\mathbf{x}$ and $\mathbf{x}_0$ in the eigenbasis $U$ of the Hessian $H = U\gamma U^\top$, which we denote by $\mathbf{v} = U^\top \mathbf{x}$ and $\mathbf{v}_0$, respectively. In that space, the solution is

$$\mathbf{v} = (\Lambda + \gamma I)^{-1} \Lambda \mathbf{v}_0 \quad \text{or} \quad v_i = \frac{\lambda_i}{\lambda_i + \gamma},$$

which means that directions with $\lambda_i \gg 0$, i.e. high curvature or strong sensitivity, stay as they are while directions with small $\lambda_i$, i.e. low curvature or "invariance", get diminished. How strongly they are diminished depends on the Lagrange multiplier $\gamma$, or, equivalently, the norm constraint.

In our analysis, we optimized the pixels based on the activations of layer `conv-3-1` – the co-trained layer – for all five models *MTL-Monkey*, *Baseline*, *Oracle*, *MTL-Oracle* and *MTL-Shuffled*. We found SGD with a learning rate of 5 to be most suitable to reconstruct from all the MTL models, and the Adam optimizer with a learning rate of 0.01 best for the Baseline and Oracle models. We always optimized for 8000 steps per image and model for each norm constraint.

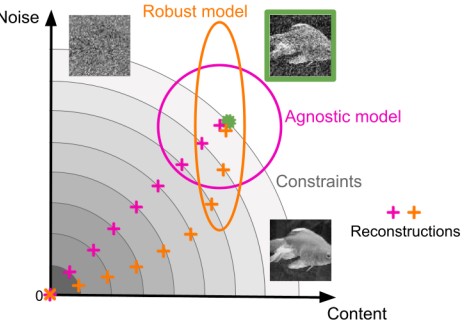

Figure 4: An illustration of the reconstruction process. We reconstruct a noise corrupted target image (green) from a given model under a resource constraint (gray circles). An agnostic model (dark pink) would be, by definition, equally sensitive to: The image content and noise. In contrast, a robust model (orange) would be more sensitive to content and less to noise. Resource constraints that do not allow for a full reconstruction force the optimization to put more image power towards directions to which the model is more sensitive.

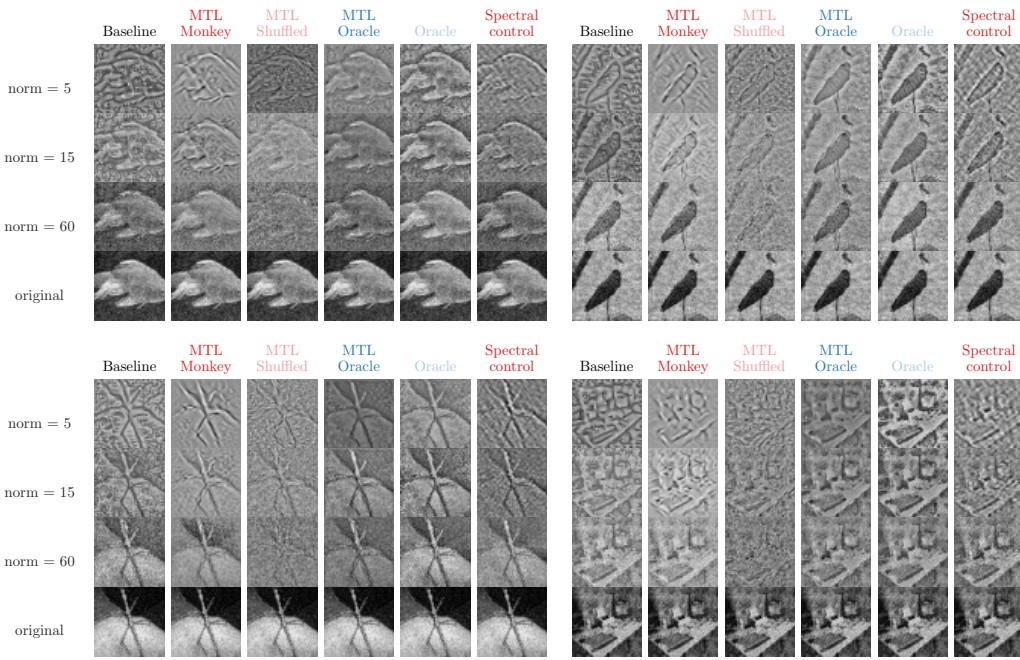

Figure 5: Reconstruction examples from all the 5 models of 4 noisy images (see last row for original images) with Gaussian noise of severity level 2, under 3 norm constraints: 5, 15 and 60. See main text for details on the spectral control.

**Reconstructions show qualitative differences between models**   Reconstructions from test images distorted with Gaussian noise under three different norm constraints qualitatively show that the models are sensitive to different features (Figure 5). When looking at a mid-range norm constraint (norm=15), where we expected the difference between the models to be the largest (see Figure 4), we found that the Baseline model seems to be sensitive to distortions present in the original image. This manifests itself in a stronger noise component in the background of the image reconstructed from the Baseline model compared to other models. The robust networks (MTL-Monkey, Oracle and MTL-Oracle), on the other hand, were less sensitive to these perturbations and exhibited more content structure for that norm constraint (see appendix for more reconstructions). However, when comparing our MTL-Monkey model with the other robust networks, we noticed that the Oracle and MTL-Oracle models generally preserved the original image content as much as possible, while reconstructions from the MTL-Monkey model seem to put slightly more emphasis on edges and object boundaries.

**MTL-Monkey's sensitivities cannot be fully explained by frequency filtering**   One simple mechanism that would make a network more robust against noise types with high-frequency perturbations would be to change the frequency sensitivity towards low-pass components. To assess whether this might be the case for the MTL-Monkey network, we used a simple "spectral control", where we transferred the Fourier amplitude spectrum of the reconstructed image to the original noisy image. This enforces the norm constraint on the original image, and exactly matches the frequency content. If the MTL-Monkey model were simply changing the frequency sensitivity, we would expect the reconstruction and the spectral control to match. However, this does not seem to be the case. The spectral control exhibited more noise in the background and showed less edge enhancing compared to the MTL-Monkey reconstruction. Thus, while changed frequency sensitivity might play a role, there might be additional effects that lead to more noise suppression and more edge enhancement.

**MTL-Monkey exhibits increased sensitivity to salient image regions**   Our constrained reconstruction analysis exposes image content that is relevant to recreate the responses of a particular layer. In this final section, we try to characterize this content in terms of known perceptual mechanisms of mammalian vision. Motivated by the potential role of V1 in bottom-up saliency [25], we specifically investigated the relation between salient regions of an image and the regions that get emphasized in

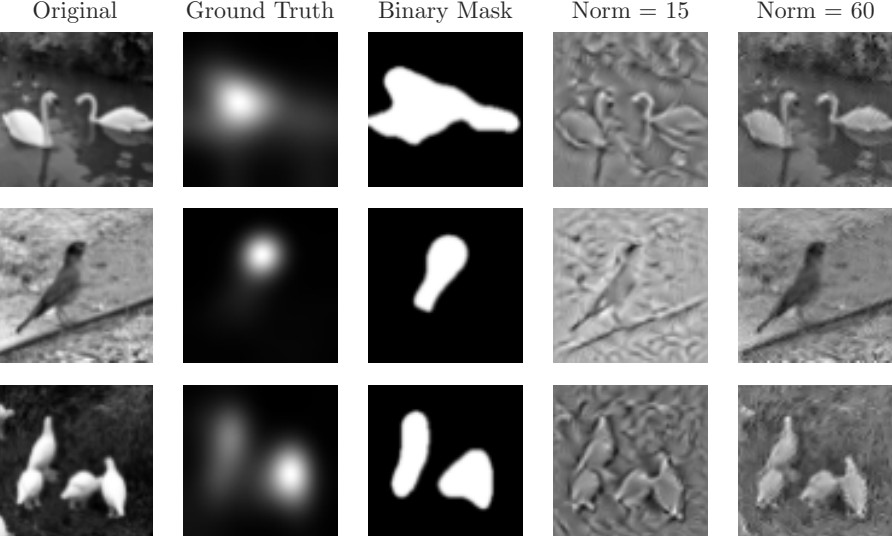

| Original | Ground Truth | Binary Mask | Norm = 15 | Norm = 60 |

Figure 6: Examples from the ImageNet images used for testing our neural saliency hypothesis in the MTL-Monkey model. In addition to the original image, we show the DeepGaze predicted saliency map and the resulting binarized mask for computing the norm ratio as well as the MTL-Monkey reconstructions with norm constraints (15 and 60).

the reconstructions from different models. Salient regions of an image often include boundaries and shapes of central objects [25, chapter 5]. It could be possible that focusing on more salient regions of an image improves the robustness of a model to corruptions as a correlation between shape-like feature detection and robustness has been reported before [10, 32]. From our qualitative observation of the reconstruction examples (Figure 5), we noticed that the reconstructions from the MTL-Monkey model seem to often enhance the central object in a scene by emphasizing edges and boundaries compared to the reconstructions from our Oracle models. To investigate whether the MTL-Monkey model is more sensitive to salient regions compared to other models, we collected 70 undistorted images with structured background from ImageNet. We used grayscale versions of these images as targets for the reconstruction from our MTL-Monkey, Oracle, and Baseline models. Additionally, we used the DeepGaze II model [34] to predict the saliency maps of these images. To define a binary saliency mask, we took the predicted density map, sorted all resulting pixel values in a descending order, and selected all pixels up to a cumulative sum of 0.7 as "salient" (see section A). Afterwards, we resized the target image and the binarized mask to 64x64 pixels, which is the standard size used for our models (see Figure 6). To quantify how much contrast is spent on the salient region salient($I$) compared to the entire image $I$, we computed the ratio $\varrho$ of the norm of the salient region against the full image's norm:

$$\varrho(I) = \frac{\|\text{salient}(I)\|_2^2}{\|I\|_2^2}.$$

We then computed the difference $\varrho\left(I_r\right) - \varrho\left(I_o\right)$ between the norm ratio of each reconstructed image $I_r$ and the original image $I_o$. To put the values on a common scale, we normalized this difference by its maximally achievable value $1 - \varrho\left(I_o\right)$ [inspired by 12]:

$$\bar{\varrho}_r = \frac{\varrho\left(I_r\right) - \varrho\left(I_o\right)}{1 - \varrho\left(I_o\right)}. \tag{1}$$

Our results show that the MTL-Monkey is more sensitive to salient regions than the Oracle and Baseline models across images under low to mid-range norm constraints (Figure 7). And as the norm approaches the norm of the full image, the ratios for both models become more equal to those of the MTL-Monkey model, as expected (right). In comparison to the Baseline and Oracle models, the spectral control seems to be closer to the diagonal. However, in most cases the MTL-Monkey model emphasized the salient regions more strongly, supporting our previous observation that frequency

filtering cannot fully explain the sensitivities of the co-trained model (see section 4). We want to stress that our analysis is purely correlational at this point: Robust MTL monkey models seem to be more sensitive to salient image content than other –robust and non-robust– models. However, if the focus on salient features turns out to be causal, it would open up the possibility that the Oracle models and the MTL-Monkey model are robust due to different reasons.

## 5    Related work

A number of previous studies also transferred useful inductive biases from biological to artificial neural networks. For instance, Arai et al. [35] utilized neural data –recorded from the superior colliculus (SC) in monkeys– to regularize the network's hidden layers for predicting saccadic eye movements accurately and improving the model's generalization abilities. Moreover, Rezai et al. [36] used the prediction of responses from the primate middle temporal area (MT) to train the lower layers of a deep CNN for visual odometry via multi-task learning, which is the main approach in our work. Although Arai et al. [35] and Rezai et al. [36] leverage the idea of co-training, their main focus is on neuroscientific modeling in contrast to our work. We directly focus on improving machine learning models, which is more in line with other related examples from the literature. Fong et al. [21] used fMRI data of the human brain's activity to guide the training of neural networks on a classification task, obtaining general performance improvements for CNNs. Other works used neural data from animals to regularize the training of artificial networks on image classification, such as in Li et al. [22], through representational distance learning (RDL, [37]). Li and colleagues used mouse V1 to regularize CNNs towards a more brain-like representation. They used the CIFAR-10 and CIFAR-100 datasets, and evaluated robustness on Gaussian noise corruptions. Similar to our findings, they observed an increase in robustness when comparing their network to the Baseline while the model regularized by the shuffled neural data did not yield similar benefits. They also observed increased robustness to adversarial attacks. We extend their work by using a larger set of 14 distinct types of corruption to evaluate the robustness on tiny ImageNet. Federer et al. [23] used neural data from the primary visual cortex of macaque monkeys for regularization, and reported an improvement in test accuracy and an increased robustness to label corruption.

Finally, previous work, summarized by Yamins and DiCarlo [16], found a strong correlation between how well a neural network performs on a classification task and how well it predicts neural responses. Recent work by Dapello et al. [33] also found that robust networks tend to be better at predicting V1 responses in macaque monkeys than non-robust models. They also reported a correlation between the brain-likeness on the representational level of a network and robustness, without explicitly training on neural data, as the model architecture was hand-crafted to emulate V1. In this work, we show similar

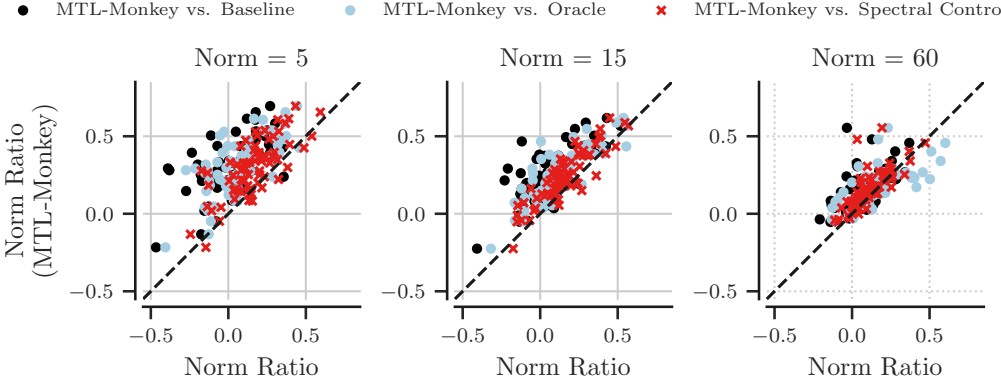

Figure 7: Each datapoint corresponds to one ImageNet image, for which we compute the normalized norm ratio (equation 1) of the reconstructed image from the MTL-Monkey model (y-axis) compared to the Oracle, Baseline or spectral control model in blue, black or red respectively (x-axis). The panels show varying norm constraints as the basis of the reconstructions. For all panels, if the datapoint is above the diagonal, it means that the salient regions in the corresponding image are more emphasized by our MTL-Monkey model than the model associated with the particular marker style.

results on the representational level by directly co-training on neural data while using an established image classification architecture (see sections 2 and 3). Therefore, we believe that our approach of learning a neurally informed representation from data is more flexible and readily generalizable in comparison to Dapello et al. [33]. For example, extending our MTL approach to higher brain areas is straightforward to do, whereas that is not immediately clear with the approach from Dapello et al. [33]. Finally, we think that Dapello et al. [33] supports our results about transferring properties of V1 into our network, especially that the similar behavior on the ImageNet-C test set, with "weather" corruptions also having a weaker performance than the rest, supports the validity of our results.

## 6 Discussion and conclusion

In this work, we show a successful transfer of robustness properties via multi-task learning on neural data and object classification. Our findings are generally consistent with prior works on inductive bias transfer from the brain, and constitutes a first test of the neural co-training hypothesis [5] for improving the robustness of neural networks. Furthermore, we are the first, to the best of our knowledge, to go a step further by introducing a novel attribution method to understand what makes our neurally co-trained model robust. Through that analysis, we find that our MTL monkey model is more sensitive to salient regions of an image compared to other models. Since V1 has been implicated in bottom-up saliency before [25], it could be that our finding might be connected to this computational property of V1 neurons. In that respect, our results are consistent with the V1 saliency hypothesis, which has been already supported by several studies in the literature [38–41].

In this work, our aim was to learn robustness from a system that is known to be robust against most corruptions – the mammalian visual system [9]. Data augmentation with image corruptions during training –as in our Oracle model– is a simple and effective baseline for robust networks [7, 42]. However, the effectiveness is limited when it comes to unseen corruptions [9], although some noise types might generalize when calibrated carefully [42]. Thus, to get a truly robust network, one would need to anticipate every possible corruption to include them in training, which is obviously intractable. Notably, our MTL approach achieves better generalization without modifying the network's input and without the additional overhead of a noise generator, and we hope that further improvements can eventually replace data corruption. Although our models are still far behind the human visual system in terms of generalization, our work is a conceptual step towards bridging the gap between artificial and biological intelligence. This could be a deciding factor in helping the reliability and generalization capabilities of computer vision. In addition, our findings might help to get a better understanding of the computational role of V1, such as its role in bottom-up saliency. A promising future direction is to include higher brain areas for neural co-training, inspired by Kietzmann et al. [43]. They trained a network for object classification with RDL on neural dynamics of multiple visual areas –including higher ones– and showed that recurrent architectures achieve better test classification performance than feedforward architectures with additional self-connections (ramping feedforward architectures). Thus, using higher areas for neural co-training while potentially relying on recurrence will presumably yield stronger robustness against more complex distortions, and when combined with our analyses, it could improve our understanding of the functional role of these areas as well.[3]

Our work represents fundamental research into the link between biological and artificial vision. Since this direction is at an early stage, the risk of misuse or unethical use of our results is present but not larger than in other fundamental investigations of the principles of vision. Our data is collected through animal experiments which are currently the only method to get high numbers of single unit responses across brain regions. All data coming from monkeys complied with the approved protocol of local authorities (see appendix). A key advantage of the type of data we used is that it is suited for a wide range of analyses beyond this paper. Therefore, our paper simply improves the scientific yield of animal recordings. Furthermore, our approach highlights that it is not strictly necessary to record dozens of new datasets for new tasks - the model that we trained on the original neural data was powerful enough to predict responses to unseen images, and these responses, in turn, can be successfully used for multi-task learning. We do believe that a network trained on a larger amount of already existing neuronal data (possibly also from other areas) of the primate visual system can be used for many more conceivable tasks. Thus, we think that our work contributes to reducing the need for invasive animal experiments and rather encourages the use of surrogate models.

---

[3]The data and code for this work are made public here: https://github.com/sinzlab/neural_cotraining.

**Acknowledgments**

We thank Konstantin Lurz, Mohammad Bashiri, Christoph Blessing, Pawel Pierzchlewicz, and Matthias Kümmerer for helpful comments on the manuscript. The authors thank the International Max Planck Research School for Intelligent Systems (IMPRS-IS) for supporting Arne Nix and Konstantin Willeke.

**Funding Transparency Statement**

Shahd Safarani was funded by the Claussen-Simon Foundation. This work was partially supported by the Cyber Valley Research Fund (CyVy-RF-2019-01). FHS is supported by the Carl-Zeiss-Stiftung and acknowledges the support of the DFG Cluster of Excellence "Machine Learning – New Perspectives for Science", EXC 2064/1, project number 390727645. This work was supported by an AWS Machine Learning research award to FHS. Supported by the Intelligence Advanced Research Projects Activity via Department of Interior/Interior Business Center contract number D16PC00003, DP1 EY023176 Pioneer Grant (to A.S.T.) and grants from the US Department of Health & Human Services, National Institutes of Health, National Eye Institute (nos. R01 EY026927 to A.S.T. and T32 EY00252037 and T32 EY07001).

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
