# A Appendix

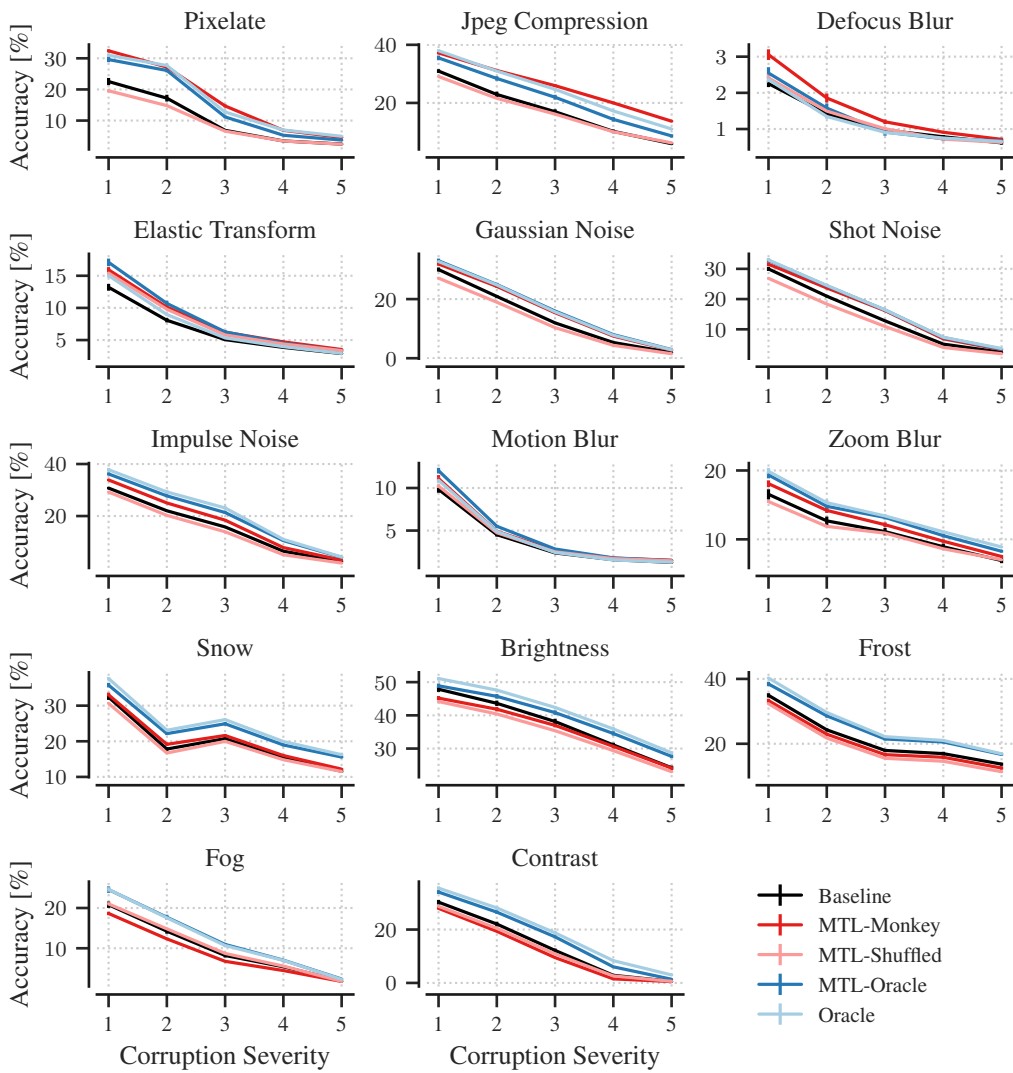

Figure 8: Our classification results on TIN-TC, showing the 14 corruption types across 5 increasing levels of severity each.

| Model | Standard Test Accuracy [%] |
|---|---|
| Baseline | 49.26 ± 1.63% |
| Oracle | 52.25 ± 0.97% |
| MTL-Oracle | 49.58 ± 1.39 % |
| MTL-Monkey | 46.28 ± 1.30% |
| MTL-Shuffled | 44.70 ± 0.61% |

Table 2: The standard (clean) accuracy performance for all our 5 models on the TinyImageNet testset, i.e. the validation set in practice, across 5 seeds per model. Here we compute the mean and standard deviation across seeds.

| Model | Robustness score |
|---|---|
| Baseline | 100% |
| MTL with real responses | 109% |
| MTL with predicted responses (MTL-Monkey) | 118% |
| MTL with shuffled predicted responses (MTL-Shuffled) | 98% |

Table 3: Comparing our MTL model co-trained on predicted neural responses –MTL-Monkey in the paper– to the MTL model co-trained directly on real monkey V1 responses. We computed the robustness score of each model after averaging the accuracies of 3 seeds per model for each corruption type in TIN-TC and normalizing against the baseline test accuracies, i.e. the baseline score is 100%. We find that we can obtain a general increase in robustness when using real neural data. However, co-training on predicted neural responses improves the robustness of the models even more. We believe, this is because the MTL-Monkey model uses the same images, i.e. tiny ImageNet images amounting to 100k images, for both tasks, namely neural prediction and image classification. On the other hand, the model co-trained with real neural data uses a much smaller set of 24k images for neural prediction (the ones which were presented in the experiment), that does not include images explicitly from tiny ImageNet but rather from ImageNet, which makes the input space of both tasks even more distinct. On top of that, we believe that denoising neural responses helps the co-training process.

**Binarization of saliency density maps:**    We did a percentile-split but based on the total "saliency mass". We applied the following approach in order to obtain a binarized mask from the corresponding density map predicted from the DeepGaze Ⅱ model. Each pixel value has a real value between 0 and 1 in the density map, representing how salient that pixel is. In order to binarize all the pixels into salient or non-salient, we only considered the pixels as salient if they contributed to the salient regions to a large extent. Therefore, we sorted out the pixel values in a descending order and then we calculated the cumulative sum of the resulting sorted array. If the pixel has a high value, i.e. large contribution to saliency in the original image, then it is by default among the first elements in the cumulative sum array. We set the threshold of whether a pixel is salient or not at a cumulative sum value of 0.7 of the "saliency mass" after trying out several thresholds and selecting one that showed (qualitatively) good binarization results on a separate set of images, that were not used in our later evaluations.

**Specifying the image resolution used for our *Monkey Predictor* model:**    The monkey V1 dataset we used for training is different from Cadena et al. [17]. In the dataset we used for this study, the neurons that were recorded are more eccentric (at approximately 4-6° from the fovea), as compared to Cadena et al. [17] with 1-3° from the fovea, which changes the size and resolution of the neuron's receptive fields. Because of the different eccentricities of the recorded neurons, the image sizes of the presented images were also changed by the experimenters, from spanning 2° visual angle to 6.7°. Our model was trained on images 4.5° in size. We cropped the images down from 6.7° as presented to the monkey to 4.5° as this sped up training time and did not result in the loss of accuracy. Because

of these changes, we ran a search for the optimal resolution in pixels per degree, which will have high performance scores in both tasks: neural prediction and image classification.

In order to achieve this, we built a new model, starting with a random VGG, and fine-tuned the early layers (up to `conv-3-1`) to predict the actual neuronal data. Then, we froze those layers and trained the rest on tiny ImageNet classification and investigated the effect of the early learnt neural representations on the classification performance (see Figure 12). We noticed that the lowest resolution values are associated with the highest classification accuracies and vice versa. This indicates that the scaling of images, which are used for neural prediction, influences the classification performance on tiny ImageNet images. Therefore, we selected a resolution of 14 pixels per degree (ppd), for which the net performance on both tasks was best. Ultimately, this is the resolution that we selected to train our neuronal prediction single-task model –*Monkey Predictor*–, which we then used for generating the responses to the TIN images.

**Supplementary Information on Data Collection**

**Animals and institutional approval**    We obtained the behavioral and electrophysiological data from two healthy, male rhesus macaque (Macaca mulatta) monkeys aged 15 and 16 years and weighing 16.4 and 9.5 kg during the time of study. The experimental procedures followed the guidelines of the NIH, and were reviewed and approved by the Baylor College of Medicine Institutional Animal Care and Use Committee (permit number: AN-4367). The Center for Comparative Medicine of Baylor College of Medicine provided veterinary care, as well as balanced nutrition and environmental enrichment. Both animals were housed individually, in large rooms (D=2' L=6' H=6'), with six other macaque monkeys, providing rich visual and olfactory interactions. All necessary surgical procedures were conducted under general anesthesia, in line with standard aseptic techniques. Analgetics were provided for seven days after each surgery, and no monkey was sacrificed following an experiment.

**Data collection**    Data was collected by non-chronic recordings using a 32-channel linear silicon probe (NeuroNexus V1x32-Edge-10mm-60-177). Under full anesthesia, custom recording chambers and headposts were implanted to enable intra-cranial recordings. Small trephinations (2mm) were made prior to the recordings over the medial primary visual cortex at eccentricities, covering a cortical area that represents visual eccentricities of 1 to 4 degrees visual angle. Recordings were performed within 2 weeks after each trephination. In all of the 32 recording sessions (15 for monkey 1, 17 for monkey 2), care was taken to drive the probe slowly into the cortex with a guide tube to minimize tissue compression.

**Stimulus presentation**    The stimuli were presented on a 16:9 HD LCD monitor, with a refresh rate of 100 Hz, with a resolution of 1920x1080 pixels. The animal subjects were placed 100 cm in front of the stimulus monitor, resulting in a viewing resolution of 63 pixels per degree visual angle. Gamma correction was applied to the monitors. In each recording session, the receptive fields were mapped via a sparse random dot stimulus. A single dot of size 0.12 degree visual angle (°) was flashed on a uniform background, with randomly changing color (black or white) and location every 30ms, while the monkey had to maintain fixation for 2 seconds on a central fixation spot. For each channel, multi-unit receptive fields were obtained with reverse correlation. The population RF was then computed by fitting a 2D Gaussian to the spike-triggered averages. To make sure that the monkeys fixated their gaze during the experiment, a custom-built camera-based eye tracking system was used. The fixation window was 0.95° around a 0.15° red fixation spot. The fixation spot was always kept at the center of the screen, with the natural images being presented at the center of the estimated population RF. The monkeys had to maintain fixation for 300 ms on the fixation spot in order to start a trial. When fixation was broken, the trial was aborted, and the next trial started when fixation was maintained again. A trial consisted of 15 images, shown back to back with no blanks in between, with 120 ms presentation time per image, which resulted in a trial duration of 1.8 s. Upon completion of a trial, the monkey was rewarded with a droplet of juice.

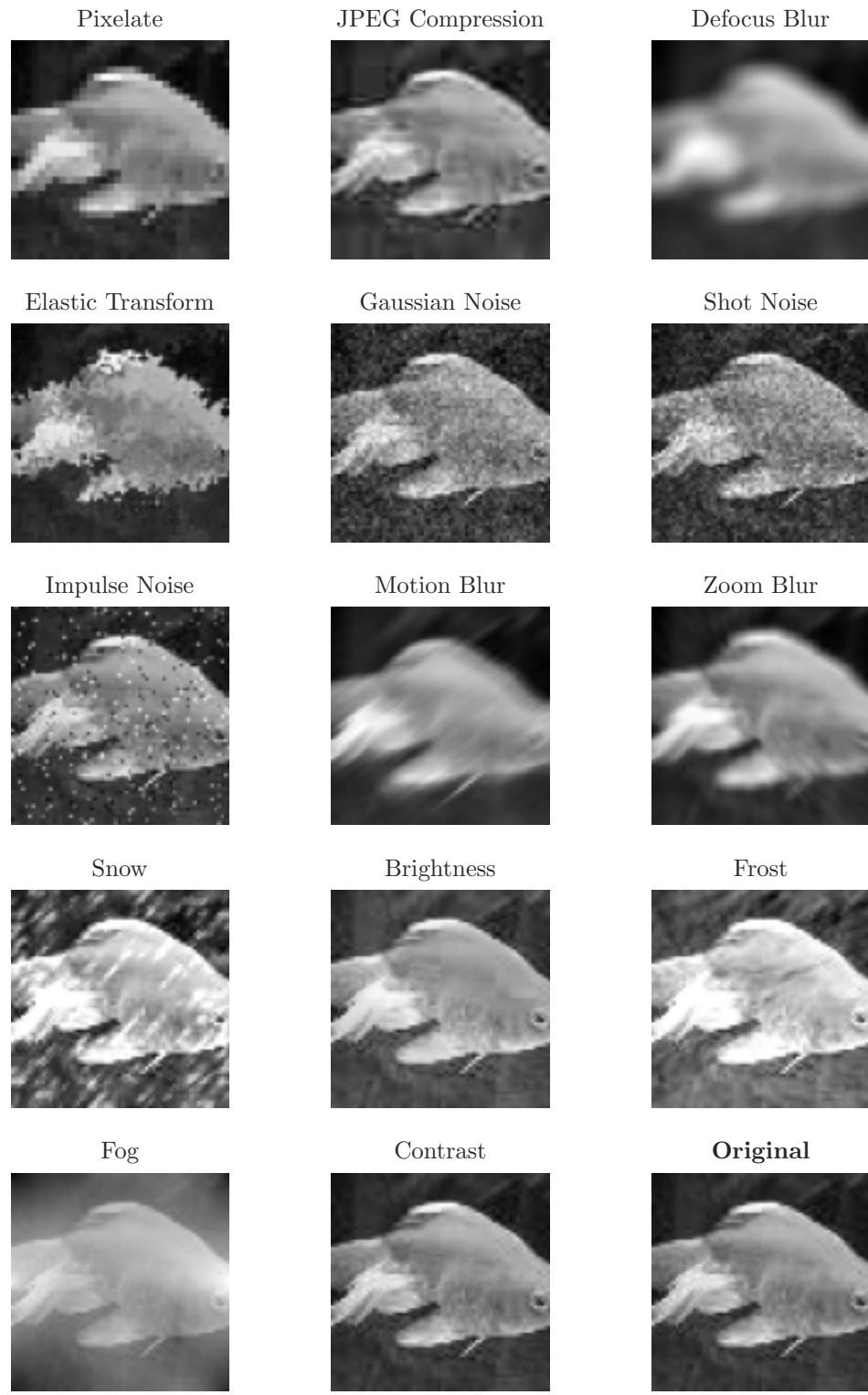

Figure 9: 14 corruption types with severity level 2 applied to an example fish image.

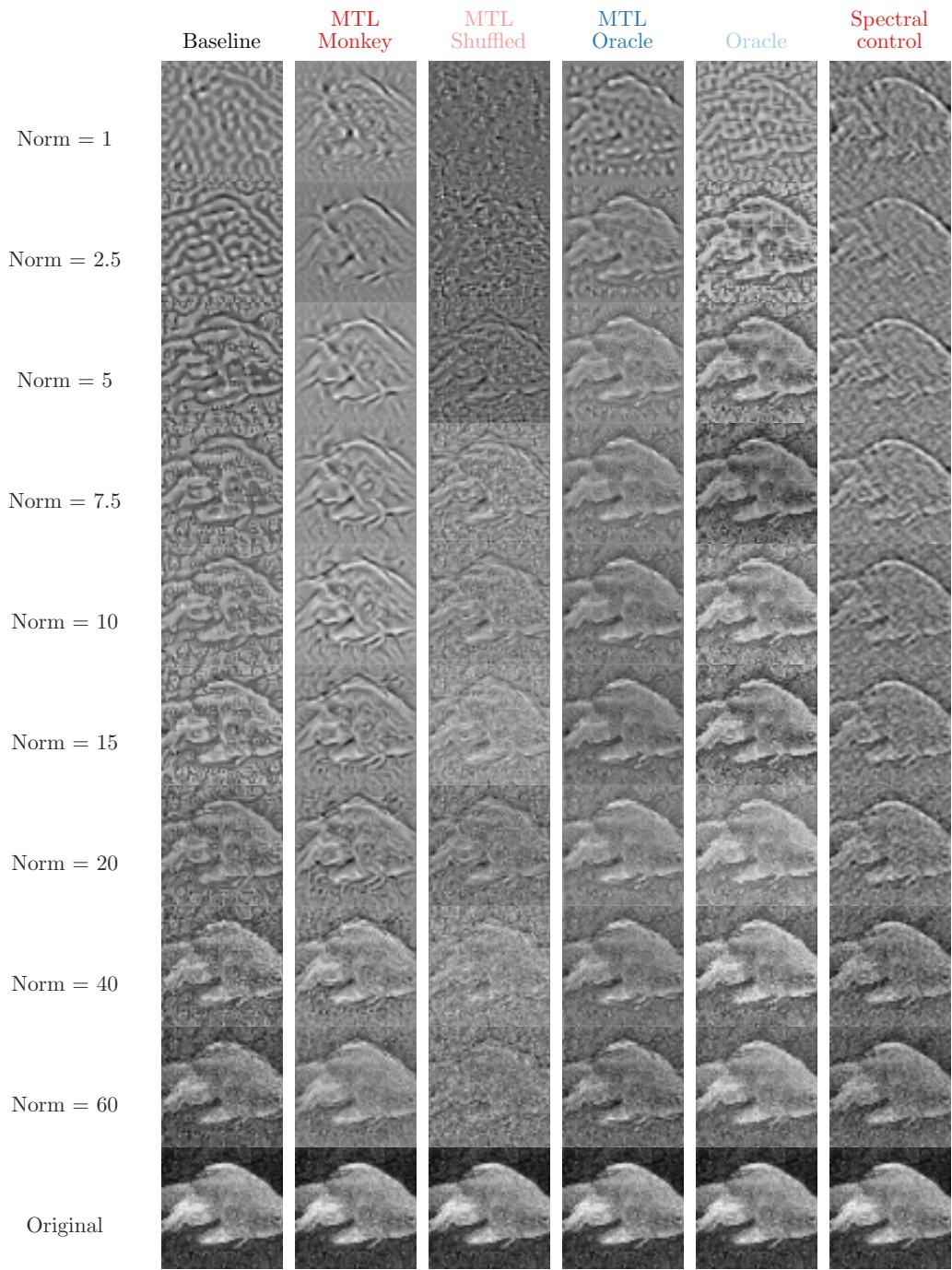

Figure 10: Reconstruction example of a noisy fish image using all 5 models with Gaussian noise of severity level 2, under all the 9 norm constraints.

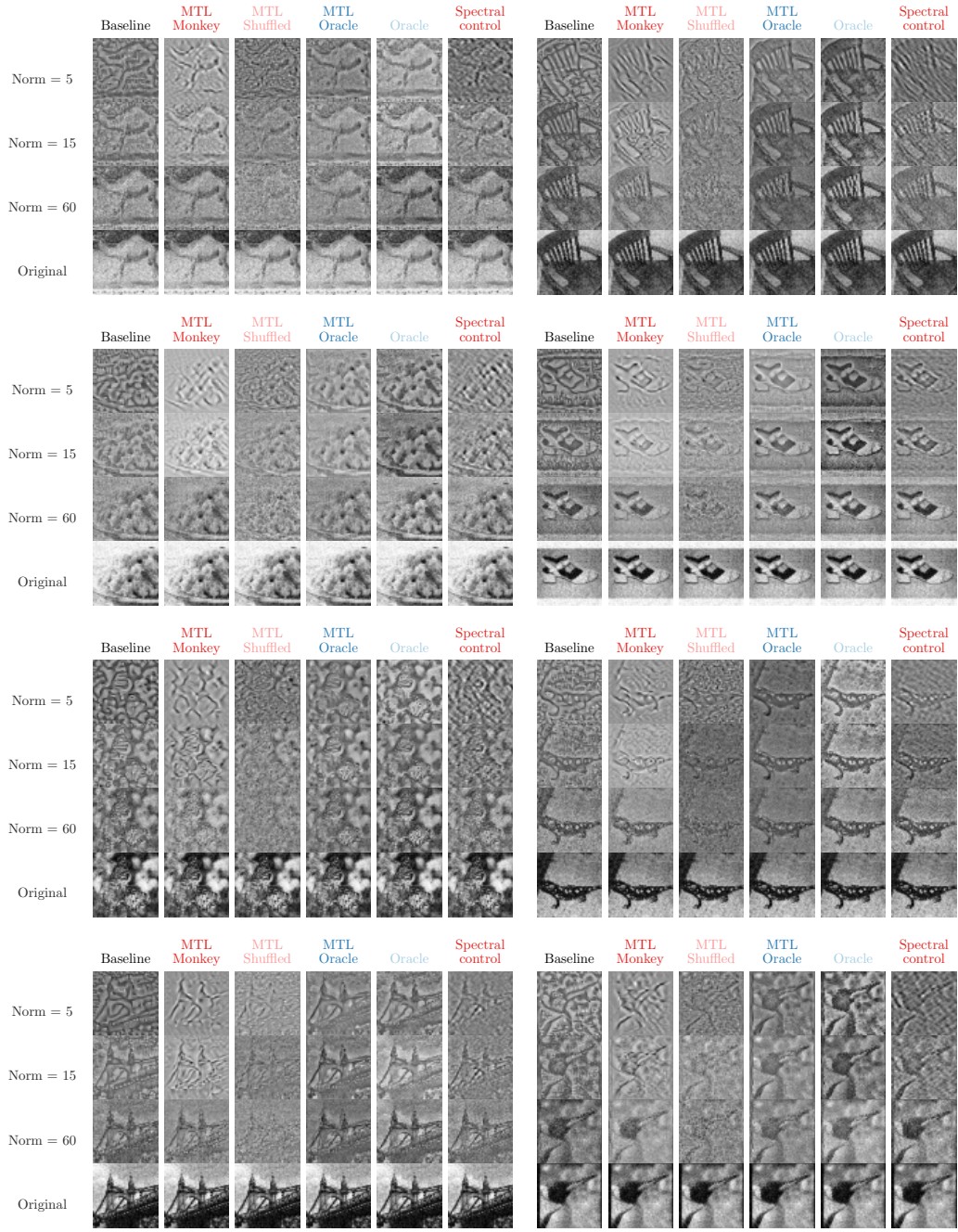

Figure 11: More reconstruction examples of 8 noisy images using all 5 models with Gaussian noise of severity level 2, under 3 norm constraints: 5, 15 and 60.

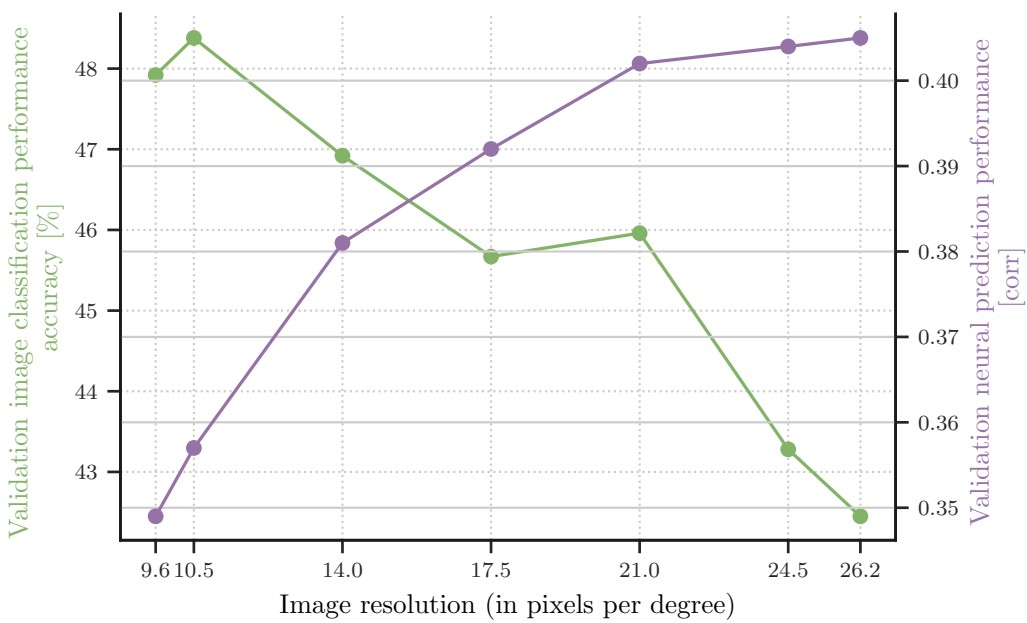

Figure 12: The performance on neural prediction of monkey V1 and its effect on the image classification performance while using different image resolution values for the input images, which were used to train the networks on monkey V1 prediction.