# OpenReview forum: "Towards robust vision by multi-task learning on monkey visual cortex"
_NeurIPS.cc/2021/Conference — NeurIPS 2021 Poster_

### Official Review · Reviewer_ji1K · 2021-07-16

**Rating:** 6
**Confidence:** 3

**Summary:**

The paper introduces a new method of regularizing deep networks with neural data. Specifically, the network is co-trained on a main task and regression of neural recordings (or at least deep-network predictions of these recordings for new images). The analysis shows that a network trained on neural responses is sensitive to different features, and encodes gaze-salient image regions more strongly, than control networks.

**Ethical Concerns:**

The main ethical issue is the use of animal data. The authors point out that the collected data is useful for a wide range of analyses beyond this one, so the study improves the usefulness of the data. If possible, this point could be expanded with examples. If the method is really effective, it seems that it could conceivably spawn an industry of new data collection for neural data in a wide variety of tasks. It would be positive to set a precedent here and release the data. There is a statement that “data coming from monkeys complied with the approved protocol of local authorities.”


**Ethics Review Area:**

["Responsible Research Practice (e.g., IRB, documentation, research ethics)"]

**Limitations And Societal Impact:**

These issues were not addressed, but I don’t think there are problematic impacts.

**Main Review:**

Originality: As the authors point out, the work is closely related to past work using representational similarity from neural recordings to regularize deep networks. The authors also mention that task/neural-data-reconstruction co-training has been suggested before (Sinz et al. 2019), but I think the idea has been in the air for some time before that, see e.g. (particularly the latter):
  Rezai, O., Jentsch, P. B., & Tripp, B. (2018). A video-driven model of response statistics in the primate middle temporal area. Neural Networks, 108, 424-444.
  Arai, K., Keller, E. L., & Edelman, J. A. (1994). Two-dimensional neural network model of the primate saccadic system. Neural Networks, 7(6–7), 1115–1135.
Nonetheless, this work seems to be the best example so far of carrying out this idea.

Quality: I think the work is sound and well developed. It could potentially be explored more thoroughly using different tasks, datasets, networks, and other measures of robustness.

Clarity: The writing is basically clear. The explanation of methodological details is compact (e.g. Gaussian readout, a recent and non-obvious approach, is only briefly mentioned), so I think it would be helpful to share the code. There are a few small points that could be further clarified, e.g.:
1) Link, reference, and description of the TINS dataset. Also explain the choice of dataset.
2) Which network was used to produce the predicted responses? I’m not sure whether line 96 refers to the model that produces predicted responses, or the site of co-training. I think the process is described on line 147, but it would help to expand the explanation around line 96.

Significance: I have three reservations about the significance:
1) Given that the approach is closely related to, e.g., Federer et al., I think it would be useful to compare the current approach to that one. Which one works better? Does each have advantages and disadvantages?
2) The main practical benefit of co-training on neural data seems to be that the network becomes nearly as robust to corruptions as a network that is trained on corrupted images. However, neural data is expensive and corrupted images are cheap, so the practical significance may be minor, or perhaps the authors could discuss it in more detail.
3) The pipeline for achieving the above benefit included corrupted images. Removing these from the pipeline, the MTL-Monkey model was notably less robust.


**Needs Ethics Review:**

Yes

**Time Spent Reviewing:**

5 hours

---

> ### Author Response · Authors · 2021-08-10
> **Author Response to Reviewer ji1K (1/2)**
>
> Dear Reviewer ji1K,
>
> Thank you for your review and feedback. We appreciate your assessment of our work as *"sound"* and *"well developed"*. We provide a point-by-point response to individual concerns below.
>
> *The idea of neural co-training has been in the air for some time before Sinz et al. 2019:*
>
> We were not aware of the two references you provided - thank you for pointing them out. Rezei et al. [1] use prediction of responses from primate middle temporal area (MT) to train the lower layers of a deep CNN for visual odometry. Arai et al. [2] also utilize neural data to aid another task, in this case, saccade prediction and test extrapolation abilities. Therefore both are relevant to our discussion and will be added to our related work section. However, as you already pointed out, our work "seems to be the best example so far of carrying out this idea". We believe this is especially true as our work focuses directly on improving machine learning models, in contrast to [1] and [2] who focus mainly on neuroscientific modeling.
>
> *This work could potentially be explored more thoroughly using different tasks, datasets, networks, and other measures of robustness.*
>
> We agree that a broader empirical study could certainly give further support to our conclusions and the neural co-training hypothesis in general. However, in the current work, given the limited space of one paper, we decided to focus on an in-depth analysis of the learned representation for one task, rather than a broad study across different tasks and models.
>
> *It would be helpful to share the code.*
>
> The code is already publicly available and will be linked in the paper upon publication.
>
> *Link, reference, and description of the TINS dataset. Also explain the choice of dataset.*
>
> Unfortunately, as it was created as part of a lecture, the tiny ImageNet (TIN) dataset does not have an official publication associated with it (there are only publications that use the dataset in some way) and the official website was taken offline recently. However, the dataset provides a challenging image recognition task that is relatively quick to train and consists of images that are closer to natural images than e.g. CIFAR100 or MNIST. Furthermore, in our case, it was particularly useful since it fits the neural data we use for our initial model which was recorded using ImageNet images as stimuli.
>
> The tiny ImageNet dataset (TIN) is a subset of ImageNet, which contains 100000 training images of 200 classes (500 for each class) downsized to 64x64 colored images. In addition, each class has 50 validation images and 50 test images. For more information, we will provide in the paper a link to the dataset description on Kaggle. Taking your feedback into consideration, we decided to add the description of the dataset to the paper, where we introduce the tiny ImageNet dataset in the Data section as well.
>
> *Which network was used to produce the predicted responses?*
>
> The network used to produce the predicted responses had the same architecture as used for all other models, namely the VGG19 with batch normalization layers. By fitting the early layers up to layer conv-3-1 to monkey V1 data directly, we were able to obtain a V1-model that we used to predict neural responses of any given input.
> As this also led to some confusion for another reviewer, we decided to rewrite the corresponding section. See also our response to reviewer KL3f for more discussion on this point.
>
> *Given that the approach is closely related to, e.g., Federer et al., I think it would be useful to compare the current approach to that one.*
>
> Please refer to our statement on our main contributions for a detailed comparison with the approach of representational distance learning that is applied for example by Federer et al. [3].
> We agree that a direct empirical comparison of the two methods would be interesting, but we decided to focus on demonstrating the feasibility of multi-task learning for transferring properties and a more detailed analysis of the resulting representations.
>
> Due to space constraints, we will continue our response in another comment below.
>
> *The main practical benefit of co-training on neural data seems to be that the network becomes nearly as robust to corruptions as a network that is trained on corrupted images. However, neural data is expensive and corrupted images are cheap, so the practical significance may be minor, or perhaps the authors could discuss it in more detail.*
>
> Data augmentation with image corruptions during training (as we used it for our Oracle model) is certainly a simple and effective baseline for robust neural networks [5,6].
> However, the effectiveness is limited when it comes to unseen corruptions [6]. Thus to get a truly robust network, one would need to anticipate every possible corruption to include them in training. This is obviously intractable, although there are promising attempts to optimize the corruption during training to help with better generalization [6].
> We go a different route and aim to learn robustness from a system that is known to be robust against most corruptions -- the mammalian visual system [8]. Notably, we do this without modifying the network’s input and without the additional overhead of a noise generator.
> Obviously, our work is only a first step and can not fully compete with data augmentation techniques yet. Nevertheless, we hope that further improvements on our approach can eventually replace data corruption, be it hand-crafted or optimized. As this is part of the core motivation for our work, we added some more emphasis on this in our paper.
>
> *The pipeline for achieving the above benefit included corrupted images. Removing these from the pipeline, the MTL-Monkey model was notably less robust.*
>
> We are not sure whether we understand your comment correctly. Thus, we would first like to stress that the MTL-Monkey model never saw any corrupted image at training time. The motivation for the MTL-Oracle model was to check if MTL can transfer robustness features from one network, i.e. the Oracle in our case, to an entirely new network at all. That’s why we used the responses to clean TIN images, which were generated from the Oracle model, instead of using the predicted responses from the V1 model for training MTL-Oracle. This way the MTL-Oracle is provided with information about corruptions only via these responses. In that sense, the MTL-Oracle model is not a real model but a control. Specifically, the MTL-Oracle model uses corrupted images in its pipeline only to train the Oracle model at first. Then, Oracle is used to generate the responses for the neural prediction objective of the MTL-Oracle model (i.e. train a robust network with data augmentation first, then simulate responses and treat them as “monkey” data in the co-training pipeline). The rest of the MTL-Oracle training follows the MTL-Monkey training in all details in order to create an "upper bound" on what could be possible through MTL in the best case.
> Nevertheless, the MTL-Monkey model did achieve improved robustness when compared to the Baseline, and is only 4% behind MTL-Oracle’s robustness, according to the following table. In this table, we recalculated the overall robustness of the given 4 models across 5 seeds per model with no bootstrapping.
>
> | Model | Baseline | MTL-Monkey | MTL-Oracle | Oracle |
> | ----------- | ----------- |----------- | ----------- |----------- |
> | Robustness score |100% | 118% | 122% | 126%|
>
> We believe that it is non-trivial that the MTL-Monkey model has a performance so close to the MTL-Oracle model. Again, the Oracle and implicitly also the MTL-Oracle model had access to the corruptions that are used on the test set during training and are thus controls. The MTL-Monkey model, on the other hand, has never seen any corrupted images anywhere in the training pipeline.

---

> ### Author Response · Authors · 2021-08-10
> **Author Response to Reviewer ji1K (2/2)**
>
> Dear Reviewer ji1K,
>
> here we continue our response from the comment above.
>
> *The authors point out that the collected data is useful for a wide range of analyses beyond this one, so the study improves the usefulness of the data. If possible, this point could be expanded with examples.*
>
> The dataset that we have used is actively being studied in fundamental neuroscientific research into the primate visual system. We’ll add a statement about ongoing research surrounding this dataset into the discussion section.
>
> *If the method is really effective, it seems that it could conceivably spawn an industry of new data collection for neural data in a wide variety of tasks. It would be positive to set a precedent here and release the data.*
>
> This is a valid ethical concern. We do plan to release our data (as we have done before with a similar dataset). However, since these experiments are costly, we have to coordinate the release with other researchers working on it before we can provide it for public download. We would like to stress one additional point: As pointed out by Reviewer KL3f, our approach highlights that it is not strictly necessary to record dozens of new datasets for new tasks - the model that we trained on the original neural data was powerful enough to predict responses to unseen images, and these responses, in turn, can be successfully used for multi-task learning. We think that it is conceivable that a network trained on a larger amount of already existing neuronal data (possibly also from other areas) of the primate visual system can be used for many more conceivable tasks. Thus, we would like to believe that our work contributes to reducing the need for invasive animal experiments and rather encourages the use of surrogate models.
>
>
> [1]: Rezai, O., Jentsch, P. B., & Tripp, B. (2018). A video-driven model of response statistics in the primate middle temporal area. Neural Networks, 108, 424-444.
> [2] Arai, K., Keller, E. L., & Edelman, J. A. (1994). Two-dimensional neural network model of the primate saccadic system. Neural Networks, 7(6–7), 1115–1135.
> [3] Callie Federer, Haoyan Xu, Alona Fyshe, and Joel Zylberberg. Improved object recognition using neural networks trained to mimic the brain’s statistical properties. Neural Networks, 131:103–114, 2020.
> [4] Fabian H Sinz, Xaq Pitkow, Jacob Reimer, Matthias Bethge, and Andreas S Tolias. Engineering a less artificial intelligence. Neuron, 103(6):967–979, 2019.
> [5] Dan Hendrycks and Thomas Dietterich. Benchmarking neural network robustness to common corruptions and perturbations. Proceedings of the International Conference on Learning Representations, 2019.
> [6] Rusak, E., Schott, L., Zimmermann, R.S., Bitterwolf, J., Bringmann, O., Bethge, M. and Brendel, W., 2020, August. A simple way to make neural networks robust against diverse image corruptions. In European Conference on Computer Vision (pp. 53-69). Springer, Cham.
> [7] R. Geirhos, C. R. M. Temme, J. Rauber, H. H. Schütt, M. Bethge, and F. A. Wichmann. Generalisation in humans and deep neural networks. In Advances in Neural Information Processing Systems 31, 2018.

---

> > ### Comment · Reviewer_ji1K · 2021-08-24
> > **Concerns partly resolved**
> >
> > Thank you for the thoughtful responses. You addressed many of my concerns and I now understand your motivation more clearly in that you see neural data as a source of robustness to unseen kinds corruptions.
> >
> > My remaining reservation about the paper is that comparison with other approaches is thin. If the effectiveness of augmentation “is limited when it comes to unseen corruptions”, it would be nice to see a clear demonstration of the superiority of the proposed method, e.g. by training with some corruptions and testing generalization to others, using the same network and task. Ref [6] in your reply also proposes a simple method that could serve as a baseline. And of course an empirical comparison to an RDL approach with the same data would be clarifying as well. I respect that you decided to focus on analysis rather than empirical comparisons. However, given that the motivation for the work is practical, I do think the paper could be much stronger with such comparisons.

---

> > > ### Author Response · Authors · 2021-08-25
> > > **Author Response to Second Comment of Reviewer ji1K**
> > >
> > > Dear reviewer ji1K,
> > >
> > > Thank you very much for increasing your evaluation of our paper and your thoughtful comments.
> > > We agree that all the experiments you suggested would be worthwhile to explore. While our focus was on the analysis of the learned representation this time, we will definitely consider your suggestions as we continue to improve and extend our approach of neural co-training.

---

> ### Author Response · Authors · 2021-08-20
> **Author Response to Ethics Reviews**
>
> Dear Reviewer ji1K,
>
> Regarding your ethical concerns, we have now devised extended statements regarding animal experiments and data collection. We kindly ask you to see our response to reviewer **igBF** for details.

---

### Official Review · Reviewer_81tw · 2021-07-16

**Rating:** 6
**Confidence:** 5

**Summary:**

This study follows a recent trend of results that use similarity to neural representations in early visual areas as a regularizer during training of deep neural networks (DNNs) for object recognition. Here the authors jointly train a DNN for image classification (grayscale version of Tiny ImageNet) and predicting neuronal responses in macaque primary visual cortex (V1), observing that this joint training procedure leads to increased robustness to common image corruptions. Finally, the authors claim that this improvement in robustness is likely to result from an increased sensitivity to salient regions of an image.

[Score updated 5 -> 6]


**Limitations And Societal Impact:**

The authors adequately addressed the limitations and potential negative societal impact of this study.

**Main Review:**

Overall the paper is well written, clear and the experiments appear to be properly done. I found the analysis on the saliency maps particularly interesting, deserving further exploration.  However, I have several concerns that negatively affect my score.


1) Main results are not entirely novel.
The authors state that the neural co-training hypothesis by Sinz et al was untested prior to this study. In my opinion, this statement is not entirely true since similar approaches have been used in the past. Specifically, Li et al NeurIPS 2019 used mouse V1 data to regularize CNNs towards more brain-like representations. This is very similar to the approach used here. While the method for introducing the regularization, the dataset (CIFAR vs Tiny ImageNet), and the species (mouse vs monkey) are different, this study is at most an extension of the existing approach, which is perfectly fine. However, as an extension it fails to make significant novel contributions. Dapello et al NeurIPS 2020 had already shown that introducing a neuroscientific model of monkey V1 leads to improved robustness to image corruptions on color ImageNet, which is a considerably larger dataset and closer to real-world applications than the one used on this study (check model trained without stochasticity for a better comparison). Furthermore, when comparing the improvements of these two studies on a corruption by corruption basis, the results are strikingly similar (improved robustness for noise, blur and digital corruptions and worse robustness for weather effects). So in summary, this study extends Li et al to a larger dataset and to predicting monkey V1, and in the process replicates findings already described in Dapello et al - macaque V1 similarity is correlated with robustness and making the early stages of the model better approximate V1 leads to improved robustness.

2) Clean accuracy missing.
Clean accuracy is only reported for the MTL-Monkey models in Figure 3C. This is problematic since the main comparisons are done using a robustness metric which normalizes accuracy to corrupted images by the clean accuracy. Also, the clean accuracy scores reported for the MTL-Monkey models appear to be surprisingly low (under 50%). If this is true, then the usefulness of this approach for improving models seems to be considerably limited.

3) Great variability in the absolute accuracy scores across corruption types.
While some perturbations show overall very large accuracies throughout the severity ranges (brightness between 30 and 50%), others have extremely low accuracies even at the lowest severity levels (defocus blur < 3%). This is very surprising and not consistent with the ImageNet-C accuracy levels. This suggests that the implementation of these corruptions for the grayscale Tiny ImageNet dataset is not appropriate and makes these results harder to interpret.

4) I am not a fan of the glass blur corruption being left out from the test dataset. Given all the compute used in this study, I fail to see how producing the glass blur perturbation for the grayscale TIN validation-set would represent a substantial increase in compute. This particular corruption affects other defense methods differently from the remaining blur corruptions and its exclusion here should be avoided.

Minor issues:

- What is the range of repetitions per images during the neurophysiological recordings (measured per neuron) for both the training and the test images? This information should be reported in the paper

- The input size of 64x64 with 14.5ppd when matching the V1 recordings means that the model’s input size in physical quantities is ~4.5deg, is this correct? This is smaller than what was considered optimal by Cadena et al 2019. Why the change?

- The authors claim that CNNs are state-of-the-art for neural response prediction of primate V1 (line 47). I would say that this statement is still somewhat under debate since Cadena et al 2019 reported only minor improvements over a very simple classical V1 model and more recently Marques et al 2021 argued that some optimized neuroscientific V1 models outperform CNNs in several V1 neural benchmarks.

[Score updated 5 -> 6]

**Time Spent Reviewing:**

7

---

> ### Author Response · Authors · 2021-08-10
> **Author Response to Reviewer 81tw (1/2)**
>
> Dear Reviewer 81tw,
>
> Thank you for reviewing our paper and for your helpful suggestions. We appreciate your assessment of our paper as &quot;well written&quot; and &quot;clear&quot; and we are glad that you found our analysis on the saliency maps &quot;particularly interesting&quot;. We are happy to address your detailed suggestions and concerns in a point-by-point response below.
>
> _1. Main results are not entirely novel. [...]_
>
> **Li et al. [5]:** Our work is, as you pointed out, related to the work of Li et al. [5]. However, we do provide several new contributions: i) using a new transfer paradigm (MTL compared to RDL matching), ii) demonstrating its benefits on a wider range and severities of image distortions, iii) using data from a different species (more closely related to humans), and iv) providing an entirely new analysis approach to understand the transferred feature representations. We kindly ask you to read our general summary above for a detailed discussion of what we consider the main novel contributions of our work.
>
> **Dapello et al. [6]:** Generally, we agree that the paper of Dapello et al. is relevant, and we updated our related work section to give an extended perspective on this. However, the work of Dapello et al. is also different in several key aspects:
>
> Firstly, Dapello et al. [6] exchange the lower layers of a given image classification architecture to _simulate_ monkey V1. The changed part of the network is not changed during training but chosen from a distribution of Gabor filters that is hand-crafted to reflect previous results of neurophysiological experiments in the literature. Additional non-linear transformations and stochasticity were also chosen to follow established findings about V1. The resulting model shows increased robustness for classification and improved &quot;brain-likeness&quot; in the lower layers, which is indeed in line with our findings. However, Dapello et al. arrive at this point from a different direction, as they hand-craft their model architecture to emulate V1, while we directly fit it to actual recordings. Secondly, we believe that our approach of learning a neurally informed representation from data is more flexible and readily generalizable. Our approach uses an established image classification architecture and extends its training procedure to fit individual neuron responses. Therefore, we learn a representation in lower layers that benefits neural prediction as well as image classification, in the hope to discover something that helps us generalize better. This does not necessarily need to be something that is already known about V1 or that could easily be hand-crafted, which led us to put a large focus on the analysis of the learned representations. Since we directly fit to neural recordings, extending this approach to higher brain areas is also straightforward to do, whereas it is not immediately clear how one would do this with Dapello et al.&#39;s approach.
>
> Finally, we do not perceive the fact that Dapello et al. arrive at similar results from a different angle as redundant, but rather see it as support for our hypothesis that the co-training approach transferred properties of V1 into our network. Especially the similar behavior on the ImageNet-C test set, with &quot;weather&quot; corruptions also having a weaker performance than the rest, supports the validity of our results. Thank you very much for pointing this out.
>
> _2. Clean accuracy missing. Clean accuracy scores reported for the MTL-Monkey models appear to be surprisingly low (under 50%). If this is true, then the usefulness of this approach for improving models seems to be considerably limited._
>
> Thanks for pointing this out. The following table includes the average clean test accuracy (%) of all our models across 5 seeds per model in addition to the standard deviation. We will add this table to the appendix as you are right that this information has to be provided in the paper.
>
> | Model               | Baseline      | Oracle        | MTL-Oracle     | MTL-Monkey    | MTL-Shuffled  |
> | ------------------- | ------------- | ------------- | -------------- | ------------- | ------------- |
> | Clean test accuracy | 49.26 ± 1.63% | 52.25 ± 0.97% | 49.58 ± 1.39 % | 46.28 ± 1.30% | 44.70 ± 0.61% |
>
> Regarding your comment that these accuracy values are low, we agree that the clean accuracy values of our models are not as high as the best scores that can be obtained on tiny ImageNet (Please note that much higher accuracies often involve training on additional data).
>
> However, our main focus here was not to achieve state-of-the-art clean accuracy performance but rather to study the benefits we can obtain in terms of robustness and generalization to out-of-distribution examples. The sub-state-of-the-art performance on clean images is likely due to our use of the VGG architecture. We chose VGG since many studies have used it for neural prediction of monkey V1 data [1]. Considering that VGG19 performs well on image classification tasks [3], we decided to use it for all our experiments to build our research on prior findings related to neural prediction. The clean performance we obtain is comparable to previously reported accuracies using VGG: When training the same VGG19 architecture we used in our paper on the colored tiny ImageNet dataset, we obtain about 56% of clean test accuracy, which was similar to the results reported by others for the same architecture [2].
>
> _3. Great variability in the absolute accuracy scores across corruption types. [...] Results not consistent with the ImageNet-C accuracy level_
>
> When we implemented our evaluation set-up, we verified that TIN-C results of a baseline model are generally consistent with IN-C [7]. To achieve this, we had to use on-the-fly corruptions, as we will discuss further below. However, we acknowledge that the accuracies for motion blur and defocus blur are surprisingly low, and agree with your assessment that the results from evaluating on these corruptions for the grayscale Tiny ImageNet dataset may not be very meaningful. Therefore, we recomputed the overall robustness of MTL-Monkey across 5 seeds with no bootstrapping and differentiated two cases: The evaluation with and without those two corruptions. We found that our observation of improved robustness via co-training still holds to a large extent, i.e. only 1.5% of drop in overall robustness, when excluding defocus and motion blur.
>
> | MTL-Monkey test on TIN-C | With defocus and motion blur | Without defocus and motion blur |
> | ------------------------ | ---------------------------- | ------------------------------- |
> | Robustness score         | 118.2%                       | 116.7%                          |
>
> Additionally, when comparing to the results of Dapello et al [7], who show IN-C results with a neuro-inspired architecture and also see an increase in robustness everywhere except for the weather category, we are confident that our results are accurate.
>
> _4. Glass blur corruption being left out from the test dataset._
>
> Thanks for directly addressing it. We would have liked to include all 15 corruptions. Unfortunately, the tiny ImageNet-C dataset that can be downloaded online, has some technical issues due to the fact that all images were first corrupted and then saved using JPEG compression, which resulted in reproducibility issues. This has been reported by several developers on GitHub and we observed similar issues (we are not allowed to provide a link to the issue here, but it should be easy to find). Therefore, we had to implement an on-the-fly corruption pipeline to avoid the problems with the jpeg compression. This way, we were able to reproduce all results related to the tiny ImageNet-C dataset. Unfortunately, we found that applying glass blur on the fly takes prohibitively long for every batch. Therefore, we decided to leave it out.
>
>
> Due to space constraints, we will continue our response in another comment below.

---

> > ### Author Response · Authors · 2021-08-10
> > **Author Response to Reviewer 81tw (2/2)**
> >
> > Dear Reviewer 81tw,
> >
> > here we continue our response from the comment above.
> >
> > On the minor issues:
> >
> > _Range of repetitions per images during the neurophysiological recordings_
> >
> > All designated training images were shown at most once per neurophysiological recording session. For our validation set, we used 20% of the training data, thus also the validation set contains images with only one presentation. The test set images were repeated 40-50 times. We will clarify this in our manuscript.
> >
> > _Explaining the change in input scale to Cadena et al 2019 [1]._
> >
> > We first want to point out that the dataset is different from Cadena and colleagues [1]. In the dataset we used for this study, the neurons that were recorded are more eccentric (at approximately 4-6° from the fovea), as compared to [1] with 1-3° from the fovea, which changes the size and resolution of the neuron&#39;s receptive fields. Because of the different eccentricities of the recorded neurons, the image sizes of the presented images were also changed by the experimenters, from spanning 2° visual angle to 6.7°. In your comment, you were entirely correct in saying that our model was trained on images ~4.5° in size. We’ve cropped the images down from 6.7°, as presented to the monkey, to ~4.5° as this sped up training time and did not result in the loss of accuracy.
> >
> > Because of these changes, we ran a search for the optimal resolution in pixels per degree, which will have high performance scores in both tasks: neuronal prediction and image classification.
> >
> > In order to achieve this, we&#39;ve built a new model, starting with a random VGG, and fine tuned the early layers (up to conv3-1) to predict the actual neuronal data. Then, we froze those layers and trained the rest on tiny ImageNet classification and investigated the effect of the early learnt neural representations on the classification performance. The following table shows the results:
> >
> > | Image resolution (in pixels per degree) | Validation neural prediction performance | Validation image classification performance |
> > | --------------------------------------- | ---------------------------------------- | ------------------------------------------- |
> > | 9.6                                     | 0.349                                    | 47.92                                       |
> > | 10.5                                    | 0.357                                    | 48.38                                       |
> > | 14.0                                    | 0.381                                    | 46.92                                       |
> > | 17.5                                    | 0.392                                    | 45.67                                       |
> > | 21.0                                    | 0.402                                    | 45.96                                       |
> > | 24.5                                    | 0.404                                    | 43.28                                       |
> > | 26.2                                    | 0.405                                    | 42.45                                       |
> >
> > As you can see, the lowest scaling factors are associated with the highest classification accuracies and vice versa. This indicates that the scaling of images, which are used for neural prediction, influences the classification performance on tiny ImageNet images. Therefore, we selected a resolution of 14 pixels per degree (ppd), for which the net performance on both tasks was best.
> >
> > Ultimately, this is the resolution that we selected to train our neuronal prediction single task model, which we then used for generating the responses to the TIN images. In the present manuscript, we report a resolution of 14.5 ppd, which is inaccurate, and should have been 14.0 ppd. We have corrected this in the revised version.
> >
> > _Claim that CNNs are state-of-the-art for neural response prediction of primate V1._
> >
> > We acknowledge that CNNs as SOTA models of primate V1 are an active topic of debate. In line with Cadena et al 2019, some recent works like Burg et al 2020 [8] do show that CNNs (goal or data-driven) still outperform (significantly) the features provided by Gabor filter banks with recifying and square nonlinearities. Even when Burg et al 2020 [8] included computations based on neurophysiological established knowledge (e.g. divisive normalization), the performance gap set by the CNN was still not fully closed. At the same time, Marques and colleagues [4] do provide convincing evidence that classical models may capture neural tuning metrics well with a competitive predictive performance on other datasets. We will thus soften our claims in our revised version of the paper, and reference Marques et al [4] accordingly.
> >
> > [1] S. A. Cadena, G. H. Denfield, E. Y. Walker, L. A. Gatys, A. S. Tolias, M. Bethge, and A. S. 371 Ecker. Deep convolutional models improve predictions of macaque V1 responses to natural 372 images. PLoS Computational Biology, 2019. doi: 10.1101/201764.
> >
> > [2]: Wu, Jiayu, Qixiang Zhang, and Guoxi Xu. &quot;Tiny imagenet challenge.&quot; Technical Report (2017).
> >
> > [3]: Karen Simonyan and Andrew Zisserman (2015). Very Deep Convolutional Networks for Large-Scale Image Recognition. In 3rd International Conference on Learning Representations, ICLR 2015, San Diego, CA, USA, May 7-9, 2015, Conference Track Proceedings.
> >
> > [4] Marques, T., Schrimpf, M., &amp; DiCarlo, J. J. (2021). Multi-scale hierarchical neural network models that bridge from single neurons in the primate primary visual cortex to object recognition behavior. Cold Spring Harbor Laboratory. https://doi.org/10.1101/2021.03.01.433495
> >
> > [5] Zhe Li, Wieland Brendel, Edgar Walker, Erick Cobos, Taliah Muhammad, Jacob Reimer, Matthias Bethge, Fabian Sinz, Zachary Pitkow, and Andreas Tolias. Learning from brains how to regularize machines. In Advances in Neural Information Processing Systems, pages 9529–9539, 2019.
> >
> > [6] Joel Dapello, Tiago Marques, Martin Schrimpf, Franziska Geiger, David Cox, and James J DiCarlo. Simulating a primary visual cortex at the front of CNNs improves robustness to image perturbations. In Advances in Neural Information Processing Systems (NeurIPS 2020), volume 33, pages 13073–13087. Curran Associates, Inc., 2020.
> >
> > [7] Rusak, E., Schott, L., Zimmermann, R.S., Bitterwolf, J., Bringmann, O., Bethge, M. and Brendel, W., 2020, August. A simple way to make neural networks robust against diverse image corruptions. In _European Conference on Computer Vision_ (pp. 53-69). Springer, Cham.
> >
> > [8] Burg, M. F., Cadena, S. A., Denfield, G. H., Walker, E. Y., Tolias, A. S., Bethge, M., &amp; Ecker, A. S. (2021). Learning divisive normalization in primary visual cortex. _PLOS Computational Biology_, _17_(6), e1009028.

---

> > > ### Comment · Reviewer_81tw · 2021-08-24
> > > **Still not convinced about the main goal of the paper**
> > >
> > > I thank the authors for the detailed answer to my comment. I still have some concerns and suggestions.
> > >
> > > Major concerns:
> > > 1 - I remain skeptical of what is the main goal of this paper. The authors show that multi-task-learning with V1 neural predictivity improves to some extent out of domain (OOD) generalization of a model. However, this improvement comes at a cost of clean accuracy and the improvements also do not appear to be that big when measured in absolute accuracy (see below). Also, there are no comparisons with other methods designed to improve OOD generalization in ImageNet-C (see Rusak et al 2020, for example), which makes it impossible to contextualize the gains with existing literature. In this sense, this study does not produce a model that advances the state-of-the-art in the defenses against common corruption. In addition, it is still not completely clear whether the resulting model is in fact a better model of V1. While the reported neural predictivities are very high, the model was trained to predict the responses to these particular neurons. It is not shown whether the model can predict held-out neurons equally well or how it compares with alternative models of V1 (Cadena et al 2019, Burg et al 2021, Dapello et al 2020). All of this would be fine, if the study adopted a completely novel approach. However, as I mentioned in my previous comment, this study feels more as an extension of Li et al 2019 and Dapello et al 2020. So in summary, while the papers shows improvements in OOD generalization and V1 neural predictivity when compared with simple baselines, there are no comparisons with alternative models, and the general approach followed by this study is not entirely original.
> > >
> > > 2 - It is very difficult to asses the improvements of the proposed model in Figure 3 due to how this metric is calculated. From what I understood, for each model, robustness is obtained by dividing the accuracy on the corruption dataset by the clean accuracy of the model, and then this value is divided by the robustness of the baseline model. This is problematic since the baseline model has a higher clean accuracy, which will result in a lower robustness even if the accuracy on the corruption dataset is the same, and inflate the normalized robustness scores. I suggest that the authors divide directly the accuracy on the dataset by the corresponding accuracy of the baseline model or that they report absolute accuracies. The fact that two normalizations are being done makes all of these analyses very convoluted.
> > >
> > > Minor concerns:
> > > 1 - Even if the provided Tiny ImageNet-C has issues with the jpeg conversion, it is not clear why the corruptions need to be done on the fly and not done a priori and saved in a lossless format.
> > > 2 - The proposed model has a very low resolution of 14 pixels per degree. This corresponds to a maximum frequency of 5 cycles per degree (Nyquist limit along the diagonal when considering the square root of 2 factor). This cuts-off the high spatial frequency neurons that are present in the V1 fovea and are important for object recognition behavior.
> > >
> > > Suggestion:
> > > - While the authors cite a paper for the the trainable observation noise parameters on the multitask training loss function, it would be nice to have some detail on the supplementary material. This is a critical aspect of the paper and the details of the model training and justification for the choice of this loss function are very thin.

---

> > > > ### Author Response · Authors · 2021-08-28
> > > > **Author Response to the Comment of Reviewer 81tw**
> > > >
> > > > Dear reviewer 81tw,
> > > >
> > > > Thanks for your thoughtful response to our rebuttal. We will further address the remaining concerns point-by-point in the following:
> > > >
> > > > **Major concerns**
> > > >
> > > > 1 - Here we would like to reiterate the goals of our paper and how we believe we have addressed them.
> > > > One goal was to investigate multitask learning as an alternative to representational distance learning (RDL) for transferring properties from recorded neural responses to an artificial neural network. As reviewer KL3f pointed out: “It isn’t obvious that this [should work], so this is an important result”. Please also refer to "Author's Summary of the Rebuttal Response" for a perspective on how the RDL method compares to our approach.
> > > >
> > > > As transferring from neural responses is a more challenging problem than transferring from other data augmentations, as it is traditionally explored in robustness research [1,2], the goal was not to reach state-of-the-art robustness, but rather to demonstrate reliable improvements for which we are confident that they are based on the actual neural recordings that we have used in our method.
> > > >
> > > > The second goal of the paper was to analyze the learned representations. We believe this aspect is quite significant and has received little attention in prior work on this topic. Therefore we dedicated the second half of our manuscript to a novel investigation towards understanding the learned representation, and made, in our view, several interesting findings in this regard. We kindly ask you to take this into account in the assessment of the novelty of our work.
> > > > We also want to clarify an issue regarding the generalizability of our neural prediction results. While it is true that our neural predictivity scores are quite high, we want to emphasize that those were calculated with respect to the denoised responses. We realize that this was phrased ambiguously in our manuscript and we will make this clearer in our descriptions.
> > > >
> > > > With regards to whether the co-trained model is a better model of V1: We do not claim that co-training results in a better model of V1. What we want to claim is that pulling the representation of the deep network towards a more V1 like representation results in increased robustness. And we showed indeed that the closer it is, the more robust it tends to be (Figure 3C).
> > > >
> > > > We did make sure, however, that our V1 predictive model that produced the denoised responses had a high predictivity score on our neuronal data, explaining 51.2% of the explainable variance (for details on the measure, see [4]), which is comparable with Cadena et al. [4], who get 52% albeit on a slightly different dataset. Furthermore, we used the identical architecture up to the readout layer as [4], followed by a more data efficient neuronal readout [5].
> > > >
> > > > Finally, regarding the issue of overfitting our model to our particular population of neurons: This certainly is a possibility. However, recent studies such as [5] show that data-driven models,  the model that produced our denoised responses, have the ability to generalize across neurons as well as stimuli. Furthermore, we also have preliminary evidence from models trained on higher areas of the macaque visual stream, that data driven representations generalize, by testing our data-driven models on sets of unseen neurons. Thus, we have reasons to believe that the data driven representation of our denoising model also generalizes.
> > > >
> > > > 2 -  We believe there has been a misunderstanding with respect to the robustness metric that we use to report our results. We’ll make sure to improve the description of that metric in our manuscript. In short, we are not  dividing by clean accuracy. The metric is actually computed along the lines that you suggested and follows the protocol of Hendrycks et al. [2]. For each corruption type, we first average over corruption-severities and then normalize this by the corresponding average baseline accuracy on the same corruption type. This results in a robustness score for a single corruption type which then can be averaged across corruptions to obtain an overall score, i.e. our robustness score (which we report in figure 3). The equation for calculating the robustness score can be found in line 135 of our manuscript.
> > > >
> > > > We apologize if this has led to any confusion and we will improve our explanation in lines 133-136 of our manuscript.
> > > > The robustness score is designed to provide a concise summary of the OOD generalization behavior. Furthermore, we also provide the absolute accuracies for three example corruptions and severities in Figure 2, as well as all other scores in Figure 8 of our supplementary material.
> > > >
> > > > **Minor concerns**
> > > >
> > > > 1 -  We implemented the on-the-fly corruptions primarily for the training of our Oracle model. In this case, the on-the-fly approach has the big advantage that the corruptions can be drawn independently for each epoch and thus extend the variety of corruptions seen in training. You are correct that it would be possible to use all corruption types in testing if we pre-computed them and saved them in a loss-less format. However, we decided to use an on-the-fly implementation instead to stay consistent with Oracle training.
> > > >
> > > > 2 -  You are correct in pointing out that our input resolution will lead to a certain amount of information loss for very high spatial frequencies (above 5 cpd). The neuronal recordings that we used for our study were recorded over the medial primary visual cortex, with eccentricities from 1.5 degree visual angle (°) to roughly 4°. Thus, neuronal responses from the area of highest acuity are not part of our dataset. Nevertheless, we did not achieve the highest possible neuronal prediction score, possibly because of the loss of information in the high spatial frequency domain. This issue does represent a technical problem when trying to balance the preferred resolutions of networks trained for object classification, and networks trained for neuronal prediction. To the best of our abilities, we tried to address that issue by finding a trade-off of optimal image resolution, to achieve a high score in both tasks (we kindly ask you to refer to the table in our response above). While it is indeed possible to select datasets (classification data and neuronal data) and their corresponding models that share the same optimal resolution, it was unfortunately out of the scope for this study due to practical limitations and dataset availability.
> > > >
> > > > **Suggestions**
> > > >
> > > > Thank you for your suggestion to extend the explanation of the multitask loss. Do we understand correctly that the introduction of the combined loss (lines 104-106) is sufficiently clear? We will do our best to expand on our motivation for this choice within the page limit, or add it to the supplementary material otherwise. To summarize our motivation:
> > > >
> > > > Cross entropy and Poisson loss, the objective functions that are commonly used for classification and neural prediction training, respectively, lie generally on different scales. One approach to combine two such losses for multitask learning would entail scaling one of them with a hyperparameter that needs to be selected by an expensive search through cross-validation. Instead, we chose to follow the example of Kendall et al. [3] and formulate both losses as their corresponding likelihood functions with learned observation noise. This is a principled way to put both objectives on the same scale and has the added advantage of automatically weighing both tasks during the optimization.
> > > >
> > > > [1] Rusak, E., Schott, L., Zimmermann, R.S., Bitterwolf, J., Bringmann, O., Bethge, M. and Brendel, W., 2020, August. A simple way to make neural networks robust against diverse image corruptions. In European Conference on Computer Vision (pp. 53-69). Springer, Cham.
> > > > [2] Dan Hendrycks and Thomas Dietterich. Benchmarking neural network robustness to common corruptions and perturbations. Proceedings of the International Conference on Learning Representations, 2019.
> > > > [3] Alex Kendall, Yarin Gal, and Roberto Cipolla. Multi-task learning using uncertainty to weigh losses for scene geometry and semantics. In Proceedings of the IEEE conference on computer vision and pattern recognition, pages 7482–7491, 2018.
> > > > [4] S. A. Cadena, G. H. Denfield, E. Y. Walker, L. A. Gatys, A. S. Tolias, M. Bethge, and A. S. 371 Ecker. Deep convolutional models improve predictions of macaque V1 responses to natural 372 images. PLoS Computational Biology, 2019. doi: 10.1101/201764.
> > > > [5] Konstantin-Klemens Lurz, Mohammad Bashiri, Konstantin Willeke, Akshay Jagadish, Eric Wang, Edgar Y. Walker, Santiago A Cadena, Taliah Muhammad, Erick Cobos, Andreas S. Tolias,  Alexander S Ecker,  and Fabian H. Sinz. Generalization in data-driven models of primary visual cortex. In International Conference on Learning Representations, 2021

---

> > > > > ### Comment · Reviewer_81tw · 2021-08-31
> > > > > **Some further clarifications**
> > > > >
> > > > > I thank the authors for the detailed responses to my questions. I have a few further clarifications regarding the model's neural predictivity and the relationship between it and robustness before deciding whether or not to change my initial score.
> > > > >
> > > > > What exactly is meant by "While it is true that our neural predictivity scores are quite high, we want to emphasize that those were calculated with respect to the denoised responses". On my first evaluation of the paper I was under the impression that the neural predictivity results in Figure 3C was a measurement of Fraction of Explainable Variance (FEV) as in Cadena et al 2019. Now, I have realized that that is not the case and the FEV for the model is "only" 51.2%. How does the analysis on Figure 3C look like when plotting Robustness against FEV?
> > > > >
> > > > > Also, since clean accuracy and robustness may be a confounding factor, how much of the correlation is due to improvements in clean accuracy and how much is done due to robustness score. To better disambiguate these two, I suggest that the authors provide  the correlation coefficients and their significance between Robustness and Clean Accuracy with neural predictivity measured as FEV in the actual neuronal data (for both the MTL-Monkey and the MTL-Shuffled models). I suggest that these plots are added in the supplementary information for completeness.

---

> > > > > > ### Author Response · Authors · 2021-09-01
> > > > > > **Author Response to the Clarification Questions of Reviewer 81tw**
> > > > > >
> > > > > > Dear reviewer 81tw,
> > > > > >
> > > > > > We thank you for your remarks and we will address your clarification questions point by point:
> > > > > >
> > > > > > 1 - We agree that this is an important analysis and we will exchange figure 3C with the neuronal predictivity for the recorded neuronal data and the FEV as evaluation measure. For this, we used our MTL-Monkey model to compute FEV scores directly on the test set of the recorded neuronal data, without retraining the readout. We averaged all scores across 5 seeds per batch ratio. Please refer to the following table for the results:
> > > > > >
> > > > > > | Clean accuracy | Robustness score | Neural prediction (FEV) | Batch ratio (Classification:Neural) |
> > > > > > |----------------|------------------|-------------------------|-------------------------------------|
> > > > > > | 46.75          | 101.5            | 23.9                    | 15:1                                |
> > > > > > | 48.4           | 111.1            | 34.6                    | 10:1                                |
> > > > > > | 47.47          | 112.2            | 43.2                    | 7:1                                 |
> > > > > > | 47.46          | 111.6            | 45.1                    | 5:1                                 |
> > > > > > | 46.96          | 111.2            | 47.0                    | 4:1                                 |
> > > > > > | 47.35          | 114.3            | 48.9                    | 3:1                                 |
> > > > > > | 46.81          | 110.4            | 51.9                    | 2:1                                 |
> > > > > > | 46.29          | 114.8            | 53.5                    | 1:1                                 |
> > > > > > | 45.52          | 116.1            | 53.8                    | 1:2                                 |
> > > > > > | 44.37          | 112.8            | 53.8                    | 1:3                                 |
> > > > > > | 44.31          | 112.2            | 53.7                    | 1:4                                 |
> > > > > > | 45.22          | 115.9            | 53.4                    | 1:5                                 |
> > > > > >
> > > > > > This table suggests that the overall picture has not changed - the robustness score does indeed increase for models with higher neural predictivity. Interestingly, the neural predictivity scores of our co-trained MTL-monkey models are equal to or even slightly surpass the model that was directly trained on the neuronal data (and that we subsequently used to generate the denoised neuronal responses), which had a score of 51.2% FEV. We thus have reasons to believe that our co-training approach results in learned representations that are able to generalize well to out of distribution data.
> > > > > > Because of time constraints, we only computed this analysis for the MTL monkey model. We will add the MTL shuffled model to our manuscript, too.
> > > > > >
> > > > > > 2 -  As we wrote in the paper, we agree that clean accuracy is a potential confounding factor for the correlation between robustness and neural performance. We repeated the analysis we describe in lines 172-174. This time, as you suggested, with FEV as the measure for neural prediction. We computed a 2-factor linear regression with FEV and clean accuracy as independent variables and robustness as a dependent variable. This resulted in an explained variance of $R^2=0.344$, with p-values (t-test) $p < 10^{-5}$ for both variables.
> > > > > > Together with our response to the first remark, these results give us further confidence in our hypothesis that brain-like representations lead to more robust models.

---

> > > > > > > ### Comment · Reviewer_81tw · 2021-09-01
> > > > > > > **Final assessment and score updated**
> > > > > > >
> > > > > > > I would like to thank the authors for the additional analysis and extensive replies to my comments.
> > > > > > >
> > > > > > > I decided to update the score to 6 and now recommend accepting the paper with some reservations. I am still not completely convinced on the novelty of the study. As I have mentioned previously, while the exact methodology used here is new, it draws many similarities with the previous studies that I referred to (Li et al 2019 and Dapello et al 2020). In addition, the lack of comparisons with other approaches makes it nearly impossible to evaluate the advantages of the proposed methodology and to contextualize it with state-of-the-art. I don't think that the paper needs to beat state-of-the-art on either robustness of V1 predictivity to be published but it is important to compare how the proposed methodology compares with existing ones (RDM similarity or using a fixed V1-like front-end adapted to the current dataset).
> > > > > > >
> > > > > > > However, while I believe that the paper could still be further improved, there are definitely some potentially interesting novel contributions to the field and hence my updated score.

---

> > > > > > > > ### Author Response · Authors · 2021-09-01
> > > > > > > > **Thanks**
> > > > > > > >
> > > > > > > > Dear reviewer 81tw,
> > > > > > > >
> > > > > > > > thanks for raising the score. We appreciate your constructive feedback which helped us improve the paper.

---

### Official Review · Reviewer_KL3f · 2021-07-19

**Rating:** 7
**Confidence:** 4

**Summary:**

This paper uses recordings from monkeys to co-train a classifier with a second head from layer conv-3-1 (of VGG-19) that predicts the cortical responses in V1 to the same pictures. The authors test this model on 14 different distortions of images and find that it is more robust than a baseline model trained only to classify the images. They show an upper-bound on robustness via a network trained on distorted images from the same distributions, and show that training on scrambled neural responses provides no advantages.

Finally, they analyze the network in two ways, qualitatively and quantitatively. First they reconstruct several input images corrupted by gaussian noise from conv-3-1 in their model and the various control networks. They find that the monkey-trained network appears to focus more on the object than the noise compared to a network trained only on the images. This is obviously subjective, so they go on to measure how much the monkey-trained network attends to salient regions (based on Deep-Gaze-2) compared to the other models, and find that it is generally more sensitive to the salient regions.

**Ethical Concerns:**

None.

**Limitations And Societal Impact:**

They mention some limitations, but I think one is that they used neural predictions rather than raw neural data. This is either a bug or a feature, depending on how you look at it. I think it should be promoted as a feature, myself, but it could be seen as a limitation.

I see no societal impacts of this work, and neither do they.


**Main Review:**

Originality: While several previous works have used neural recordings (fMRI, monkey cortex) to regularize their models, this is (I believe) the first to co-train a model on categorization of the image as well as regression to the monkey recordings. Furthermore, I don’t believe anyone has measured the robustness to such a broad array of types of distortions in a neurally-constrained model.

Quality: Everything appears technically sound. The results are strong, and there are thorough control networks, each controlling for different things, or testing different aspects. There is a baseline model that is only trained on clean images. Then there is the main model, named “MTL-monkey”, which instantiates the main idea of co-training on clean images and predicting V1 neural recordings. Actually, the “neural recordings” are from a network trained to produce them. This is done to give the MTL network a single target for each image, since the monkey recordings vary across presentations. It is unclear why this was necessary; one wonders what would happen if the MTL-monkey network had been trained directly on the data. However, (see below), this neural prediction network is used to make “pseudo-recordings” on images never shown to monkeys, expanding the dataset.

There are two “oracle” models as well. The first, simply titled the “Oracle”, is trained on a 50-50 split of clean and distorted images, using both classification and SimCLR-type representation learning between clean and distorted images at the conv-3-1 level. Now that the front end (up to conv-3-1) has been trained to be robust to distortions, the rest of the network is re-trained on clean images. This represents an upper bound on robustness where the front end of the network is what is robust to noise.

The second oracle network is a multi-task trained network, called MTL_Oracle. It is identical to the MTL-monkey model in structure, but it is trained on neural predictions from the Oracle network. To provide these predictions, the Oracle network is frozen and V1 predictions are trained from the representations at conv-3-1. These neural predictions from the Oracle network are thus trained from a front-end that has been “robustified” to distortions, and provides an upper-bound on the amount of transfer learning possible from the neural signal. Then MTL-Oracle is trained only on clean images, but the targets are from a network sensitive to image distortions. This results in a network that is nearly as good as the Oracle network, and better than the baseline network, demonstrating that transfer learning is possible in principle.

Clarity: The paper is written in a pretty confusing manner. It was not easy to extract the above description from the paper. There are a dizzying array of both single-task and MTL models. For example, the “Models” section starts on line 92. But at the end of the paragraph above that, the paper describes a single-task network that is simply trained to map from the images to the neural recordings. This network is then used to predict neural recordings for all of the TIN (Tiny Image Net) images. These predictions are then used in the following models. So, it seems that not all of the neural recordings that the MLT-monkey model is trained on are actually neural recordings, but instead are **predictions** of neural recordings. This makes sense, as it expands the data that can be used to train the neural responses, given the issue that monkey recordings are difficult, and one might not have neural recordings for all images. However, this should be made clearer in the paper. An example of where this leads to confusion is line 167, where you describe your model as being trained on “real neural data.” It isn’t, if I understand correctly.

Furthermore, the Models section doesn’t describe all of the models. Instead, it describes generically what network is used, and how it is modified to be fully-convolutional.

Another confusing part is the description of the Oracle training. This is reported in the “Results” section - it probably should be back in the “Training” section, or the “Models” section. On line 141, it describes how the Oracle network, trained on 1:1 clean:distorted images, is frozen up to layer conv-3-1 and then the rest of the network is re-trained only on clean images. But then on line 142-144, they mention further training on the frozen part using SimCLR-style training to map the distorted and clean images closer together in layer conv-3-1. If you are doing that, it isn’t frozen! So, I infer that you are describing how the 1:1 network was trained **before** it was frozen. These sentences should be rearranged. Oh, and it isn’t clear that after you have frozen the network up to layer conv-3-1, that you re-initialized the rest of VGG-19 to random weights again, i.e., trained that part from scratch. I assume that’s what you did.

It would be best to systematically describe each model, and why you created it. First, there’s the neural recording predictor, used to create single targets for each image, and targets for images that were never actually shown to monkeys. This is the “neural data” that you train the MTL model on, as well was what you trained the frozen Oracle model on to make MTL-Oracle. The reason for MTL-Oracle is to demonstrate that transfer learning is possible. It might make the paper clearer to leave out MTL-Oracle altogether, since you demonstrate transfer learning with MTL-Monkey, but that’s up to you. Also, you should describe MTL-shuffled in the methods before the results. Why did you create MTL-shuffled? Just to show that adding on a second head doesn’t help if it isn’t making correct predictions (this seems a bit silly, but ok). Because the reader encounters Figure 3 before the descriptions of all of the different models, many of the names given there are somewhat mysterious.

Under the Models section, you mention using pooling; I assume you are referring to global average pooling? If so, say so.

It’s a bit weird to say that you learn to balance the two losses, but then say you can control it with the batch ratio. Then you use a batch ratio of 1:1! It is only later that this is used in Figure 3C - it might be good to say why you have this earlier. In the batch ratio, I assume that in order to implement this, you have instances where you only collect gradients from the neural head when the ratio favors the neural recordings and vice versa? Also, in Figure 3C, you mean the regression is from neural performance to robustness, right? Not the other way around.

It would be good to have a figure showing a clean image along with the 14 distortions. We shouldn’t have to go back to the Hendrycks and Dietterich paper to find out what they look like.

I notice there are no error bars on the baseline network.

Line 172: What was the range of batch ratios used?

Line 174: You describe this as “two-factor regression” - I read that to mean you had two independent variables. What are the two factors in the regression? There’s an independent variable (neural prediction) and a dependent variable (robustness), right? How can there be a p-value for both?

lines 248-251 (sentence starting “It could be…”) I can’t parse that sentence. Please rewrite it.

Line 260: I don’t know what you mean by all pixels up to a cumulative sum of 0.7. Pixels go from 0-255. What are you summing? Why not just do a median split?

In the discussion, you mention that it would be interesting to use higher brain areas in training. I think this has already been done in Kietzmann et al,, (2019) Recurrence is required to capture the representational dynamics of the human visual system. PNAS 116(43):21854-21863. There, they used the dynamics of RSA over time to train a network to have those dynamics.

Significance: The results here are very interesting: Training a network to also predict neural responses makes it more robust. It isn’t obvious that this should be so, so this is an important result. I just wish the paper had been written more clearly.





**Time Spent Reviewing:**

4.75

---

> ### Author Response · Authors · 2021-08-10
> **Author Response to Reviewer KL3f (1/2)**
>
> Dear reviewer KL3f,
> Thank you very much for your valuable feedback. We appreciate your assessment of our work as *"thorough"* and *"technically-sound"* and our results as *"very interesting"* and *"important"*.
>
> First, we would like to address the clarity of our writing, as we understand this to be your main point of concern with our paper. To address this issue, we followed most of your suggestions and restructured the paper to improve readability. Please regard the excerpt below, where we introduce all models in a single place (what was previously the "Model" section is now titled "Architecture") to mitigate confusion, and further simplify things by summarizing the models in an overview table. Should you still find the paper unclear, we would appreciate your feedback so we can make the changes necessary to improve it.
>
> > **Models** We use a VGG-19, like we described it above, trained on grayscale TIN to serve as the *Baseline* for our experiments. To prepare our neural co-training, similar to Li et al. [21], we first trained a *Monkey Predictor* model on the image-response pairs of our recorded neural data. We then used this to predict neural responses for all input images of the TIN classification data. These predicted responses served as the basis neural dataset we used in our MTL approach. This allowed us to balance the amount of data we have for each task and it removed trial-to-trial variability in the neural data.
> Since co-training only affects the shared representation up to layer conv-3-1, we cannot expect the network to be as robust as a network where all layers are trained on data augmented with the image distortions. To explore the limits on robustness resulting from sharing lower layers only, we trained a classification model with a 1:1 mixture of clean and distorted images drawn from the pool of 14 ImageNet-C [7] corruptions (cf. Figure 9 for examples). To push the robustness to the frozen part, we added a second loss that penalizes the Euclidean distance between the outputs of layer conv-3-1 for the same image augmented with different corruptions – similar to Chen et al. [28]. We then froze all layers up to conv-3-1, re-initialized the rest, and re-trained the remaining network on clean data only. We refer to this model as the *Oracle* since it has access to the image distortions during training – unlike our MTL models.
> To demonstrate that MTL can in principle transfer robustness properties without showing distorted images in training, we generated neural responses from our Oracle model for all images of the clean TIN dataset by freezing the Oracle model and training a Gaussian readout on top of layer conv-3-1 for 10 epochs to predict V1 data. Then, we trained a model on the resulting neural responses alongside clean image classification using MTL. We call this model *MTL-Oracle*. This model also gives us a realistic “upper bound” for our MTL experiments.
> For our main experiment, we trained MTL with the neural responses generated from the Monkey Predictor model, and refer to it as *MTL-Monkey*. This model has never seen distorted images at any point. To demonstrate that MTL-Monkey has an effect beyond introducing noise into the training, we perform a control experiment which we refer to as *MTL-Shuffled*. For this, we train a model on the same neural data but with shuffled responses across images for all neurons.
>
> > | Model | Classification | Neural Prediction |
> | ----------- | ----------- | ----------- |
> | Baseline | Clean TIN | – |
> | Monkey Predictor| – | Monkey responses |
> | Oracle | Noise augmented TIN |– |
> |MTL-Oracle | Clean TIN | Oracle model responses |
> |MTL-Monkey | Clean TIN | Monkey predictor responses |
> | MTL-Shuffled | Clean TIN | Monkey predictor responses (shuffled) |
>
> In addition to the suggested changes that we cover in our rewrite, we address your detailed suggestions and concerns in a point-by-point response below.
>
> *It is unclear why [training on predicted neural responses] was necessary; one wonders what would happen if the MTL-monkey network had been trained directly on the data.*
>
> We investigated this point in preliminary analyses and compared the MTL model on predicted neural responses (referred to as MTL-Monkey in the paper) to the model co-trained directly on real monkey V1 responses. As in the paper, we computed the robustness score of each model after averaging the accuracies of tiny ImageNet-C (TIN-C) across 3 seeds per model and normalizing against the baseline test accuracies ( i.e. the score of the baseline is 100%). This leads to the following robustness scores:
>
> | Model |Baseline | MTL with real responses | MTL with predicted responses (MTL-Monkey) | MTL with shuffled predicted responses (MTL-Shuffled) |
> | ----------- | ----------- | ----------- | ----------- | ----------- |
> | Robustness score | 100% | 109% |118% | 98% |
>
> As you can see in the table, we find that we can obtain a general increase in robustness when using real neural data for co-training on image classification. However, it seems that co-training on predicted neural responses improves the robustness of the models even more. We believe that this is the case because the MTL-Monkey model uses the same images, i.e. tiny ImageNet images amounting to 100k images, for both tasks, namely neural prediction and image classification. On the other hand, the model co-trained with real neural data uses a much smaller set of 24k images for neural prediction (the ones which were presented in the experiment), that does not include images explicitly from tiny ImageNet but rather from ImageNet, which makes the input space of both tasks even more distinct. On top of that, we believe that denoising neural responses helps the co-training process.
>
> *It might make the paper clearer to leave out MTL-Oracle altogether.*
>
> We considered this suggestion but decided to leave the MTL-Oracle in the paper because it gives a more realistic "upper bound" to our MTL-Monkey model compared to the model we call  Oracle. Since only the only constraint comes through the neural data, it’s unclear how much robustness increase we can expect at all. Thus, we consider the Oracle models an important control which shows that multi-task-learning can successfully transfer features, such as robustness, to an entirely new network without the requirement for direct training on corrupted images. We hope the changes to the "Model" section also make this point more apparent.
>
> *Under the Models section, you mention using pooling; I assume you are referring to global average pooling?*
>
> This refers to a maximum-pooling operation. This is the default for the VGG architecture, and we indicate it in the legend of figure 1. However, we understand that this is somewhat imprecise in the text. The changed version of the text will make this clearer.
>
> *It’s a bit weird to say that you learn to balance the two losses, but then say you can control it with the batch ratio. [...]*
>
> Thank you very much for pointing this out. Through your input, we realized that it is actually unnecessary to introduce the batch-ratio at this point. Our changed version now addresses the batch-ratio setting (in simplified form) only at the correlation experiment (figure 3 C), which is the only point in the manuscript where we use it.
>
> *In the batch ratio, I assume that in order to implement this, you have instances where you only collect gradients from the neural head when the ratio favors the neural recordings and vice versa?*
>
> The batch-ratio is in fact about determining the number of batches sampled from each dataset which correspond to either image classification or neural prediction. For example, if we want to sample two batches from the neural dataset and one batch from tiny ImageNet, then we would sample these three batches in total and accumulate the gradients meanwhile. After the sampling is finished, we would backpropagate the error signals throughout the network, starting from the classification and neural heads (at once) back to the first layer in the network.
>
> *Also, in Figure 3C, you mean the regression is from neural performance to robustness, right? Not the other way around.*
>
> Yes, the regression is computed from neural performance to robustness (i.e. robustness is the prediction target). Thank you for pointing this out. We corrected it in the figure description.
>
> *It would be good to have a figure showing a clean image along with the 14 distortions.*
>
> We do provide such a figure. Unfortunately due to space-constraints, we had to move it to the appendix (Figure 9). We will make sure to refer to this figure more clearly in the main text.
>
> *I notice there are no error bars on the baseline network.*
>
> On figure 3-A and 3-B, the error bars for the baseline are by definition zero since we compute the robustness scores of all models normalized against the baseline. On figure 2, we do show the error bars for the baseline curve, but it is barely visible due to the small variance of the accuracy values for that model.
>
> *Line 172: What was the range of batch ratios used?*
>
> Let’s assume T the number of batches used for image classification, while N is related to neural prediction. Then, we define the neural batch-ratio to be N:T. The batch ratios used in Figure 3-C are the following: [1:15, 1:10, 1:7, 1:5, 1:4, 1:3, 1:2, 1:1, 2:1, 3:1, 4:1, 5:1]. Upon your feedback, we now include this information in a clear manner in our revised version of the paper as well, when we address Figure 3-C and describe it in the paper.
>
>
> Due to space constraints, we will continue our response in another comment below.

---

> ### Author Response · Authors · 2021-08-10
> **Author Response to Reviewer KL3f (2/2)**
>
> Dear Reviewer KL3f,
>
> here we continue our response from the comment above.
>
> *Line 174: You describe this as “two-factor regression” - I read that to mean you had two independent variables. What are the two factors in the regression? There’s an independent variable (neural prediction) and a dependent variable (robustness), right? How can there be a p-value for both?*
>
> There are indeed two independent variables in our regression, which are the clean test performance for image classification and neural prediction. In addition, there is one dependent variable, which is robustness. Thus we obtain a p-value for clean image classification and neural prediction, respectively. Our goal is to analyze how the performance of the co-trained model on each task contributes to the robustness of the model or, to paraphrase, we investigate how much of the robustness results from better clean accuracy and how much is influenced by the ability to do neural prediction.
> We are sorry if the formulation in the paper was confusing. We adapted it to make this clearer.
>
> *Lines 248-251 (sentence starting “It could be…”) I can’t parse that sentence. Please rewrite it.*
>
> We will address this point in our revised version of the paper by rewriting that sentence and separating it into two sentences. Thank you for pointing this out.
>
> *Line 260: I don’t know what you mean by all pixels up to a cumulative sum of 0.7. Pixels go from 0-255. What are you summing? Why not just do a median split?*
>
> What we did is basically a percentile-split but based on the total “saliency mass”. We applied the following approach in order to obtain a binarized mask from the corresponding density map predicted from DeepGaze ||. Each pixel value has a real value between 0 and 1 in the density map, representing how salient that pixel is. In order to binarize all the pixels into salient or non-salient, we only considered the pixels as salient if they contributed to the salient regions to a large extent. Therefore, we sorted out the pixel values in a descending order and then we calculated the cumulative sum of the resulting sorted array. If the pixel has a high value, i.e. large contribution to saliency in the original image, then it is supposed to be among the first elements in the cumulative sum array. We set the threshold of whether a pixel is salient or not at a cumulative sum value of 0.7 of the “saliency mass” after trying out several thresholds and selecting one that showed (qualitatively) good binarization results on images not used in the later evaluation.
>
> *In the discussion, you mention that it would be interesting to use higher brain areas in training. I think this has already been done in Kietzmann et al, (2019) Recurrence is required to capture the representational dynamics of the human visual system. PNAS 116(43):21854-21863. There, they used the dynamics of RSA over time to train a network to have those dynamics.*
>
> It is true that in the mentioned paper, Kietzmann et al. [1] study the modeling of neural dynamics in higher brain areas. Also, they test their models on object recognition, suggesting that neuroscience-related findings could be potentially inspiring for computer vision applications. However, their main focus is on the benefits they could bring to neuroscience. Overall, it is very much related to our work and we agree that it is important to consider this work for future experiments. We will mention this in our related work section and would like to thank you for pointing this out.
>
> *[...] they used neural predictions rather than raw neural data. This is either a bug or a feature, depending on how you look at it. I think it should be promoted as a feature, myself, but it could be seen as a limitation.*
>
> So far we have not put much emphasis on the fact that from the limited neural data we recorded, we were able to produce synthetic responses for the entire tiny ImageNet training set.
> We agree with you that this should be promoted as a benefit of our approach that minimizes the need for animal experiments and allows for more flexibility in experimental design. We will add a statement on this to our paper. Thank you very much for this suggestion.
>
> [1] Kietzmann TC, Spoerer CJ, Sörensen LK, Cichy RM, Hauk O, Kriegeskorte N. Recurrence is required to capture the representational dynamics of the human visual system. Proceedings of the National Academy of Sciences. 2019 Oct 22;116(43):21854-63.

---

### Review · Ethics_Reviewer_zwaG · 2021-08-12

**Recommendation:**

I encourage the authors to include additional details of the data collection (in the appendix would be fine) as well as if the project went through an IRB. I'm am unsure of the level of documentation typical in neuroscience papers, but the norms of that community would be a useful standard upon which to assess the detail provided in this paper.

**Ethical Issues:**

Yes

**Ethics Review:**

As far as I can tell the main ethical concern of this work relates too the dataset of neutral recordings. The authors provide minimal details of the protocols of collection. The dataset is also collected from animal neural recordings posing potential ethical concerns.

---

> ### Author Response · Authors · 2021-08-14
> **How to provide more details without breaking anonymity?**
>
> Dear reviewer zwaG,
>
> please see our response/question to reviewer igBF.

---

### Review · Ethics_Reviewer_igBF · 2021-08-13

**Recommendation:**

The issues in this work mostly stem from lack of detail about the protocol and requirements that the authors went through to collect neural data from the Monkeys. This issue can be easily addressed with an additional paragraph to the document and further details to the appendix.

**Ethics Review:**

This work studies and explores the extrapolation capabilities of multi-task learning models trained on image classification and prediction of neural responses from Monkeys. The key issue here is the use of this animal data in the experiments. The paper says, "All data coming from monkeys complied with the approved protocol of local authorities", which one of the reviewers pointed out. I think this statement is not sufficient to justify the overall protocol in this work, and additional detail needs to be provided.  I looked at the appendix and this detail is not provided either. Lastly, the IRB portion, which I presume the data collection study went through is answered with N/A. Having said all of the above, the paper does mention that protocols were observed, but it just doesn't provide enough information on this front.

---

> ### Author Response · Authors · 2021-08-14
> **How to provide more details without breaking anonymity**
>
> Dear reviewers igBF and zwaG,
>
> thanks for your comments. As the experimental protocol has already been used to record data for published studies before, we are confident that we can address all points you brought up in your ethical reviews. However, we are not sure how to do that without breaking anonymity. If we reveal more details about the protocol and the authorities who approved it, you could easily identify us. That's also the reason why we kept it short in the paper. If you or the area chair could comment on this, we are happy to post more details here and include them in the paper later.

---

> ### Author Response · Authors · 2021-08-20
> **Response to Ethics Review**
>
> Dear reviewers **igBF** and **zwaG**,
>
> We have now devised fully anonymized statements about the institutional approval of our animal experiments, as well as more details regarding data collection and processing. We will also add these statements to the appendix of our article, in order to extend our brief descriptions in the main text regarding data collection, processing, and stimulus protocol.
>
>
> **Ethics statement**
>
> We obtained the behavioral and electrophysiological data from two healthy, male rhesus macaque (Macaca mulatta) monkeys aged 15 and 16 years and weighing 16.4 and 9.5 kg during the time of study. The experimental procedures followed the guidelines of the national institute for the care and use of laboratory animals, and were reviewed and approved by the **anonymous US institute’s** committee on animal care and use (permit number: AN-4367).
> The **anonymous US institute’s** center for comparative medicine provided veterinary care, as well as balanced nutrition and environmental enrichment. Both animals were housed individually, in large rooms (D=2' L=6' H=6'), with six other macaque monkeys, providing rich visual and olfactory interactions. All necessary surgical procedures were conducted under general anesthesia, in line with standard aseptic techniques. Analgetics were provided for seven days after each surgery, and no monkey was sacrificed following an experiment.
>
>
>
> **Data collection**
>
> Data was collected by non-chronic recordings using a 32-channel linear silicon probe (NeuroNexus V1x32-Edge-10mm-60-177). Under full anesthesia, custom recording chambers and headposts were implanted to enable intra-cranial recordings. Small trephinations (2mm) were made prior to the recordings over the medial primary visual cortex at eccentricities, covering a cortical area that represents visual eccentricities of 1 to 4 degrees visual angle. Recordings were performed within 2 weeks after each trephination. In all of the 32 recording sessions (15 for monkey 1, 17 for monkey 2), care was taken to drive the probe slowly into the cortex with a guide tube to minimize tissue compression.
>
>
> **Stimulus presentation**
>
> The stimuli were presented on a 16:9 HD LCD monitor, with a refresh rate of 100 Hz, with a resolution of 1920x1080 pixels. The animal subjects were placed 100 cm in front of the stimulus monitor, resulting in a viewing resolution of 63 pixels per degree visual angle. Gamma correction was applied to the monitors. In each recording session, the receptive fields were mapped via a sparse random dot stimulus. A single dot of size 0.12 degree visual angle (°) was flashed on a uniform background, with randomly changing color (black or white) and location every 30ms, while the monkey had to maintain fixation for 2 seconds on a central fixation spot. For each channel, multi-unit receptive fields were obtained with reverse correlation. The population RF was then computed by fitting a 2D Gaussian to the spike-triggered averages.
> To make sure that the monkeys fixated their gaze during the experiment, a custom-built camera-based eye tracking system was used. The fixation window was 0.95° around a 0.15° red fixation spot. The fixation spot was always kept at the center of the screen, with the natural images being presented at the center of the estimated population RF. The monkeys had to maintain fixation for 300 ms on the fixation spot in order to start a trial. When fixation was broken, the trial was aborted, and the next trial started when fixation was maintained again. A trial consisted of 15 images, shown back to back with no blanks in between, with 120 ms presentation time per image, which resulted in a trial duration of 1.8 s. Upon completion of a trial, the monkey was rewarded with a droplet of juice.

---

### Author Response · Authors · 2021-08-10
**Author's Summary of the Rebuttal Response**

We would like to thank all reviewers for their valuable feedback and we very much appreciate their assessment of our work as *"sound"* (**R-KL3f**, **R-ji1K**), *"well developed"* (**R-ji1K**), and *"well written"* with an *"interesting"* analysis (**R-81tw**) and experiments with *"strong"* results (**R-KL3f**).

Main concerns of the reviewers and how we addressed them:

Reviewer **KL3f** was mainly concerned with the clarity of writing, especially in introducing the variety of models we use for our evaluation. We are very thankful for this remark and the helpful suggestions on how to improve this in the manuscript. Taking all suggestions into account, we decided to rename the "Model" paragraph to "Architecture" as this better reflects its content. Furthermore, we added a new "Model" paragraph and moved the introduction and motivation of all models to this point. An additional table detailing each model's training data for neural prediction and image classification respectively is meant to give a quick overview to help distinguish the six models (also to clarify confusion for **R-ji1K**).

A main concern of reviewer **81tw** was the novelty of our main results. Reviewer **ji1K** had related concerns regarding the significance of our co-training method in terms of practical applicability and in comparison to related work. We addressed both concerns in detail in our respective responses. However, we believe that we can clarify most of this by expanding on what we believe are two main contributions of this paper:
1. We employ a new approach to introduce the neural data through co-training. Prior methods [1,2] generally use representational distance learning (RDL [3]) to introduce the knowledge from neural recordings into the network. RDL tries to match the representational dissimilarity matrices (RDM) of a network's hidden representation and the recorded activity from the visual cortex of an animal. In contrast, we try to predict the raw activity of individual neurons. To do this, we introduce an additional projection in the form of a Gaussian readout head [8] to predict the neural activity from any layer in a standard image recognition network.
We believe that this makes our approach significantly distinct from the established way of RDL, and justifies an investigation. Additionally, as reviewer **KL3f** rightfully pointed out, it is not self-evident that this would work.
2. Although others have shown generalization effects similar to what we see for our method, understanding this effect, to the best of our knowledge, has not been investigated so far. Thus we dedicate the second half of our paper entirely to our analysis, in an effort to shed light on this question. Using our novel reconstruction method, we find that reconstructions from different models show qualitative differences (lines 220-231), the MTL-Monkey model’s sensitivities behavior cannot be fully explained by frequency filtering (lines 232-242) and finally that the MTL-Monkey exhibits increased sensitivity to salient image regions (lines 243-277).

Additional points we addressed:

- Related work (**R-KL3f** , **R-ji1K**, **R-81tw**): extended discussion of [1,2,4] and comment on [5,6,7]
- Discussion on how our method compares to data augmentation techniques (**R-ji1K**)
- Comment on ethical concerns regarding animal experiments (**R-ji1K**) and positive aspects of our synthetic neural data (**R-KL3f**)
- Addition of missing clean accuracy values for MTL models (**R-81tw**)
- Addressing variability in absolute accuracy scores for some corruption types by comparing to robustness scores without the outlier corruptions (**R-81tw**)
- Showing robustness scores for MTL with raw (instead of predicted) neural data ( **R-KL3f** )


[1] Zhe Li, Wieland Brendel, Edgar Walker, Erick Cobos, Taliah Muhammad, Jacob Reimer, Matthias Bethge, Fabian Sinz, Zachary Pitkow, and Andreas Tolias. Learning from brains how to regularize machines. In Advances in Neural Information Processing Systems, pages  9529–9539, 2019.
[2] Callie Federer, Haoyan Xu, Alona Fyshe, and Joel Zylberberg. Improved object recognition using neural networks trained to mimic the brain’s statistical properties. Neural Networks, 131:103–114, 2020.
[3] Patrick McClure and Nikolaus Kriegeskorte. Representational distance learning for deep neural networks. Frontiers in computational neuroscience, 10:131, 2016.
[4] Joel Dapello, Tiago Marques, Martin Schrimpf, Franziska Geiger, David Cox, and James J DiCarlo. Simulating a primary visual cortex at the front of CNNs improves robustness to image perturbations. In Advances in Neural Information Processing Systems (NeurIPS 2020), volume 33, pages 13073–13087. Curran Associates, Inc., 2020.
[5]: Rezai, O., Jentsch, P. B., & Tripp, B. (2018). A video-driven model of response statistics in the primate middle temporal area. Neural Networks, 108, 424-444.
[6] Arai, K., Keller, E. L., & Edelman, J. A. (1994). Two-dimensional neural network model of the primate saccadic system. Neural Networks, 7(6–7), 1115–1135.
[7] Kietzmann TC, Spoerer CJ, Sörensen LK, Cichy RM, Hauk O, Kriegeskorte N. Recurrence is required to capture the representational dynamics of the human visual system. Proceedings of the National Academy of Sciences. 2019 Oct 22;116(43):21854-63.
[8] Konstantin-Klemens Lurz, Mohammad Bashiri, Konstantin Willeke, Akshay Jagadish, Eric Wang, Edgar Y. Walker, Santiago A Cadena, Taliah Muhammad, Erick Cobos, Andreas S. Tolias,  Alexander S Ecker,  and Fabian H. Sinz. Generalization in data-driven models of primary visual cortex. In International Conference on Learning Representations, 2021

---

### Decision · Program_Chairs · 2021-09-27

**Decision:**

Accept (Poster)

**Comment:**

This paper shows that co-training on primate neural data improves the robustness of CNNs.  There were some questions about the relevance of the work with respect to other previous work, but this was handled well by the authors and the reviewers were satisfied by those clarifications (which should also be worked into the paper).  There were some ethical concerns raised, but these also were handled well.  In general, I believe the review process worked well here, the paper is stronger because of it, and should be accepted.